# Impacts of observation frequency on proximity contact data and modeled transmission dynamics

**Weicheng Qian**[1☯¤]*, **Kevin Gordon Stanley**[1☯¤], **Nathaniel David Osgood**[1,2,3☯¤]

**1** Department of Computer Science, University of Saskatchewan, Saskatoon, SK, Canada, **2** Department of Community Health and Epidemiology, University of Saskatchewan, Saskatoon, SK, Canada, **3** Bioengineering Division, University of Saskatchewan, Saskatoon, SK, Canada

☯ These authors contributed equally to this work.
¤ Current address: Department of Computer Science, University of Saskatchewan, Saskatoon, SK, Canada
* weicheng.qian@usask.ca

**Data Availability Statement:** Unfortunately, we cannot make participants' data (SHEDs) accessible to the public because the Research Ethics Board (REB) of the University of Saskatchewan has concerns. Such concerns include that no safety

## Abstract

Transmission of many communicable diseases depends on proximity contacts among humans. Modeling the dynamics of proximity contacts can help determine whether an outbreak is likely to trigger an epidemic. While the advent of commodity mobile devices has eased the collection of proximity contact data, battery capacity and associated costs impose tradeoffs between the observation frequency and scanning duration used for contact detection. The choice of observation frequency should depend on the characteristics of a particular pathogen and accompanying disease. We downsampled data from five contact network studies, each measuring participant-participant contact every 5 minutes for durations of four or more weeks. These studies included a total of 284 participants and exhibited different community structures. We found that for epidemiological models employing high-resolution proximity data, both the observation method and observation frequency configured to collect proximity data impact the simulation results. This impact is subject to the population's characteristics as well as pathogen infectiousness. By comparing the performance of two observation methods, we found that in most cases, half-hourly Bluetooth discovery for one minute can collect proximity data that allows agent-based transmission models to produce a reasonable estimation of the attack rate, but more frequent Bluetooth discovery is preferred to model individual infection risks or for highly transmissible pathogens. Our findings inform the empirical basis for guidelines to inform data collection that is both efficient and effective.

## Author summary

Close-proximity human contacts play a fundamental role in the spread of many diseases. While the advent of commodity mobile devices have eased the collection of contact time series, battery capacity and associated costs impose tradeoffs between the frequency and scanning duration used for contact detection and participant experience and adherence. To understand the impact of the frequency with which human contact networks are

measures can ensure participants' privacy for public access because SHEDs' proximate contact data is longitudinal (over a month) and with high granularity (collected every 5 minutes). Additionally, in the informed consent form that the participants signed, researchers committed to only providing data in aggregate to ensure privacy. Therefore, to disclose the full dataset would violate that commitment. Interested researchers can request access to the data through requests to the Research Ethics Board (REB) of the University of Saskatchewan. The latest contact information of REB can be found on the website of the University of Saskatchewan. At the moment, the webpage of REB is at https://research.usask.ca/rei/researchers/ethics/human-ethics.php with the following contacts: Nick Reymond (Research Ethics Specialist) at 306-966-2084 or nick.reymond@usask.ca Melanie Bayly (Research Ethics Coordinator) at 306-966-4390 or melanie.bayly@usask.ca The model and source code used to produce the results and analyses presented in this manuscript are available from GitHub Git repository: https://github.com/Quar/ObsFreqImpact.

**Funding:** This work was supported by the Natural Sciences and Engineering Research Council of Canada (RGPIN-2020-04866 to KGS; RGPIN-2017-04647 to NDO). The funder had no role in study design, data collection and analysis, decision to publish, or preparation of the manuscript.

**Competing interests:** The authors have declared that no competing interests exist.

observed on the accuracy of network reconstruction and simulated attack rate, we down-sampled data from five high-velocity contact network studies, each measuring participant contacts every 5 minutes over at least four weeks. Results from infection transmission models parameterized by contact networks reconstructed from successively downsampled contact information revealed that the model-predicted attack rate and the per-realization variability in predicted attack rate varies markedly by pathogen and network structure. For some pathogens across multiple studies, downsampling contact rates imparts pronounced inaccuracies in model-predicted attack rate, compared to what is predicted with highest-velocity contact data. Our findings can inform design of data collection studies that are both efficient and effective, and may aid understanding of contact networks beyond the current collection limit.

This is a *PLOS Computational Biology* Methods paper.

## Introduction

Despite a century of advances, the burden of contagious diseases remains troublingly high. In the context of growing rates of drug resistance and virus mutations, development patterns which elevate human contact with vectors and animal disease reservoirs, and the capacity of infections to be disseminated via historically growing rates of global travel, the potential burden of infectious disease is historically high. From the shocking worldwide death toll from SARS-CoV-2 [1–4], to Middle East respiratory syndrome coronavirus (MERS-CoV), to Ebola in central Africa [5, 6], to the burden of endemic tuberculosis worldwide and in indigenous communities [7, 8], to the lost productivity due to seasonal flu [9–11] and the common cold [12, 13], and the resurgent patterns of childhood communicable diseases [14–16], contagious disease continues to impose a heavy adverse impact on society. This impact has driven substantial and ongoing research into the transmission, population spread, treatment, and prevention of common viral and bacterial pathogens [14, 17, 18]. For the past century, dynamic models of communicable diseases have served as a key tool in the understanding, prevention and control of communicable disease. A central element of such models is a representation of contact patterns between hosts, transmission, and the natural history of infection within a host [19, 20].

Close-proximity human contact networks constitute a key mechanism in the spread of communicable diseases [21–23]. Together with pathogen-specific parameters, high-fidelity representations of such contact networks within transmission models [22] can enable a much higher resolution view of the process of a disease spreading than is possible with the random mixing assumptions required in compartmental models within the traditional susceptible-infectious-recovered (SIR) family [19, 20, 24]. Such a view can support real-time identification early of outbreaks and an estimation of the attack rate, as well as retrospective evaluation and assessment of improved effectiveness of altered vaccine schedules, aid in planning of interventions such as outbreak response immunization [25], public health orders and quarantine, and support assessment of the impact of the scope, speed, and breadth of contact tracing [26]. Transmission models structured with a detailed contact network aid inferencing of population-scale effects from individual-level behavior of infections by enabling characterization of

the transmission of contagious diseases over the close-proximity contacts shaping outbreak dynamics [22, 27].

The ubiquity of smartphones with their rich complement of sensors, and emergence of wearable proximity-detection device have enriched data collection systems [28–33]. Automatic contact tracing apps using Bluetooth low-energy [34] have allowed researchers to collect contact information whose self-reporting would be burdensome [35, 36], and likely infeasible due to limited awareness of contacts [37]. As envisioned by some observers [22], the growing availability of proximity contact data in high-resolution has further encouraged analytics taking empirical data of proximate contacts into transmission modeling [29, 38–41]. Salathé *et al.* [29] pioneered collecting high-resolution proximity contact data with mote sensors, and taking such high-resolution data into a transmission model to analyze influenza outbreaks. Wymant *et al.* [42] investigated the impact of the National Health Service (NHS) COVID-19 app for England and Wales and estimated that increasing app uptake could reduce the number of cases.

Despite the increasing scale of computing power in the form of expanding storage capacity and accessible high-performance computing, we still struggle to collect, store, and process individual-level contact data sufficient to parameterize a longitudinal transmission model with even a municipal-scale population. When configuring smartphones to collect proximity contact data, a sensing regime with sampling frequencies on the scale of minutes notably elevates power consumption, risking adverse impacts on study recruitment and adherence. Such impacts are of particular concern among low-socioeconomic status populations who are subject to elevated risks of communicable disease transmission due to crowding and other risk factors [43–45].

In light of such technology constraints, past contributions [23, 28, 46] have argued that a clear understanding of the sensing regime is required—a sensing regime schedules short periods to turn sensors on for scanning throughout an experiment. The proximity contact data in our study are derived from Bluetooth discovery records, and the Bluetooth discovery is performed at the first minute of each duty cycle, where duty cycles are consecutive periods of identical length. The reciprocal of the duty cycle interval is referred to as the observation frequency, and the observation frequency is in inverse relationship with the inter-observation interval. This study investigated how varying sensing regimes impacts captured proximity contact data and the results of an empirical contact empowered transmission model (ECTM). Specifically, we sought to investigate the following three questions:

- How does the structure of the inferred contact network skew as the observation frequency of Bluetooth discovery reduces?

- How do the results of a transmission model when taking proximity contact data collected at a reduced observation frequency deviate from taking proximity contact data collected at a baseline frequency (the highest frequency among our scenarios)?

- Under which disease/pathogen and community structure contexts may observation frequency be reduced, and to what extent, without undermining confidence in conclusions?

We addressed these questions by analyzing proximity contacts derived from downsampled contact data collected from participant smartphones in five high-resolution human contact network studies. Each study has an effective duration of four or more weeks, and includes at least 30 participants, yielding a total of 284 participants across all studies. Close-proximity contact data were collected approximately every 5 minutes by smartphone-based Bluetooth handshakes. We analyzed how network structure changed as observation frequency is reduced.

To study the impact of downsampling on the model-estimated attack rate and individual infection risks, we provided downsampled contact data to an SEIR agent-based simulation model for 12 different transmissible diseases/pathogens. Using findings at the baseline resolution (involving sampling every 5 minutes) as the reference, we found that the bias-variability of the attack rate shifted as observation frequency was reduced. Our findings further demonstrate that in terms of both variability and bias, the magnitude of the impact of reducing observation frequency is both disease and community specific. Specifically, for diseases with low basic reproductive number, such as Middle East Respiratory Syndrome (MERS), simulation results with respect to both attack rate and individual infection risk were relatively insensitive to observation frequency. On the other hand, pathogens such as *Bordetella pertussis* showed a marked dependence on sampling frequency. Maintaining a higher observation frequency notably turns out to be more important in denser communities. Finally, we found that individual infection risk varied according to which edges of contact network served as parts of transmission chains within a given simulation.

## Data sources

This study drew contact data from five high-velocity microcontact data sets each with a month or longer duration, employing the Saskatchewan Human Ethology Datasets (SHED) 1, 2, 7, 8, and 9 [31, 47, 48]. These SHED data sets employed the iEpi system and its successor Ethica Data [31, 49] to collect longitudinal data via smartphone-based sensors, including with respect to the battery level, charging state, Bluetooth, Wi-Fi, GPS, accelerometer, magnetometer, in addition to pre- and post-surveys. Only the Bluetooth discovering records and battery level records were used in this research. It is important to emphasize that the SHED datasets, though sharing high acquisition velocity and a duration of a month or greater, exhibit notable heterogeneity in the characteristics of the participant population and—by extension—the network structures. SHED1 and SHED2, represent "closer" communities, composed of graduate students and staff from the Department of Computer Science from University of Saskatchewan, with SHED1 having the majority of its participants coming from two research laboratories. In contrast, SHED7, SHED8, and SHED9 recruited undergraduate students from across the University of Saskatchewan through a social sciences study pool, representing a more diverse and "diffuse" community. Descriptive information of these SHED data sets, such as aggregate distributions, can be found in the S1 Appendix and code repository.

All SHED studies' participants were volunteers. No experimental manipulations were conducted during data collection. The studies did not undertake stratified sampling as to ethnicity, grade, or gender. The study did not proscribe participation by those connected with the department or research laboratories involved, and the study team informed colleagues in labs and the Department of Computer Science first. Awareness of the potential study involvement can be assumed to have spread across social networks. For SHED1 and SHED2, participants were provided with a pre-configured Android phone that they carried in conjunction with any other personal mobile device. By contrast, participants used their own phones for SHED7, SHED8, and SHED9. Although for these three studies, both Android and iPhone users were welcome, because Bluetooth beaconing did not work reliably on iPhone due to security settings, iPhone users were removed from the analysis and all participants reported here were Android users.

## Contact data collection method

Data collection for Bluetooth contacts and battery levels on both iEpi and Ethica Data apps equipped smartphones occurs within discontinuous epochs. Study periods (consecutive days

spanning at least one month) were divided into 5-minute (exactly for SHED1 and SHED2, and approximate intervals for SHED7, 8, 9) duty cycles. Within each duty cycle, battery levels were recorded as long as the apps were running, and Bluetooth scan was enabled during the first minute of each duty cycle. Phones were discoverable while scanning for nearby discoverable devices.

## Methods

We synthesized collections of proximity contact data with varying sensing regimes by down-sampling from a baseline. The impact of varying sensing regimes are measured on two types of findings: those regarding network structure, and those involving population-wide disease spread. For the network analyses, we compared network structure with successive levels of downsampling and interpreted the results in terms of classical network models [50–52]. For the simulation analyses, we used an individual-level Susceptible-Exposed-Infectious-Recovered (SEIR) model [53], with reconstructed contact networks using 12 distinct common communicable diseases/pathogens (flu, SARS, fifth, pertussis, measles, chickenpox, MERS, diphtheria, COVID-19 wild type, COVID-19 Alpha variant, COVID-19 Beta variant, COVID-19 Delta variant). We investigated how downsampling (decrements in observation frequency) impacts findings regarding the attack rates, individual infection risks, and outbreak timing from simulation outputs, by employing two distinct downsampling methods named `Snapshot` and `Upperbound`. For every combination of choices from downsampling methods, sampling rates, communicable diseases, and studies, the contact network for that study induced by that downsampling rate was derived and analyzed, and simulations conducted using those networks were analyzed.

### Ethics statement

SHEDs data collection and analysis was conducted under written approval BEH-14–203, from the University of Saskatchewan Human Behavioral Ethics Review Board. Written informed consent was obtained from the participants.

### Downsampling approach

We assume that the behavior of close-proximity contacts is time-varying and denoted by an undirected graph $G_t = (V_t, E_t)$, with vertices representing participants and edges denoting pairs of participants that exhibit close-proximity contact at time $t$. We assume that, given a sufficiently small temporal quantum $\xi_0$ (for example, one second), the state of our close-proximity contacts can be considered constant across each such time quantum without significant loss of precision, meaning our analysis only considers dynamics over a unidimensional lattice with spacing $\xi_0$. This leads to proximity contacts evolving over time as a series of undirected graphs $G_{t_0}, \ t_0 \in \xi_0 \mathbb{N} \subset \mathbb{R}$, where $t_0$ projects the discrete-time index onto a real-world clock. We denote proximity contacts among participants $V$ at time $t$ as $G_t, \ t \in \mathbb{N}$. Downsampling according to a heuristic is essentially aggregating $\{G_t\}, t \in [t_i, t_j)$, which can be considered as a coding problem [54, 55].

Because the baseline frequency of longitudinal data obtained is approximately every 5 minutes, the original sampling of close-proximity contact network is a series of $\{G_t\}, t \in [0, T)$, where $t$ has the unit of minute and $T$ is the effective length of a study in minutes. After post-processing, $t$ represents an integer index representing the minute associated with the observation, where minute 0 corresponds to the first minute of the first day of the study. For convenience, we rephrase the sample time as a period rather than a specific point,

$\{G_{t_i}\}$, $t_i \in [i\xi, (i+1)\xi)$, $i = 0, 1, 2, \cdots$, where $\xi = 5$ is the (expected) duty cycle interval and $1/\xi$ is the observation frequency for our original data, and is referred to as the baseline frequency.

A further consideration relates to data availability. Such availability is affected by many factors, including—but not limited to—participants opting to "snooze" the sensor data recording during a private period, cases where the operating system temporarily evicts the data collection app from memory due to resource shortages, or—especially for the case of SHED7 and 8—due to misaligned duty cycles reflecting system scheduling. After aggregation, each sample $G_t = (V_{t_i}, E_{t_i})$ is an unweighted unidirected simple graph which can be represented as a $(0, 1)$-adjacency matrix. This adjacency matrix is symmetrical and each of its element $a_{ij} \in \{0, 1\}$ indicates individual $v_i$ and $v_j$ have a contact ($a_{ij} = 1$) or no contact ($a_{ij} = 0$).

We considered two downsampling strategies: A physically realizable sampling strategy (named `Snapshot`), and a theoretical upper-bound (named `Upperbound`). Snapshot periodically samples a snapshot of the current contacts in place at that time, thereby providing a simulated answer to the question "what if we sampled less frequently?". The `Upperbound` downsampling strategy instead records all contacts throughout the downsampling interval, and reports those as applying at the sampling time. It instead answers the question "What would be the impact of these same contacts, if they were to change less frequently?" `Upperbound` provides an oracle which maintains all contacts during the period regardless of whether the downsampled schedule would have measured them.

**Snapshot.**    The `Snapshot` downsampling method is conceptually straightforward: for each downsampling period $[i\xi', (i+1)\xi')$, $i = 0, 1, 2, \cdots$, we choose the first available sample index $G_{\tilde{t}_i}$, $\tilde{t}_i \in [i\xi', (i+1)\xi')$. This results in subsampling $\{G_{\tilde{t}_i}\}$, $\tilde{t}_i \in [i\xi', (i+1)\xi')$. If a contact occurred during the specific duty cycle captured by that index, it will be reflected within the sampled record. Snapshot simulates the effect of selecting a longer duty cycle for measurement, including the loss of contacts due to undersampling.

**Upperbound.**    In contrast to `Snapshot`, we sought to investigate the impact of a theoretical downsampling method, which could provide a sample summary that included information drawn from throughout that interval. Specifically, we considered the union $G_{\tilde{t}_i}$ for $\{G_{t_i}\}$, $t_i \in [i\xi, (i+1)\xi)$, $i = 0, 1, 2, \cdots$, where the union, in general for any discrete set $j \in \mathbb{N}$, is defined as $\bigcup_{j \in J} G_j = \bigcup_{j \in J} (V_j, E_j) = (\bigcup_{j \in J} V_j, \bigcup_{j \in J} E_j)$. This downsampling mechanism serves to conserve all pairwise contacts which are observed at any time during a downsampling interval. `Upperbound` cannot practically be deployed in data collection using the most common sensors used for proximity detection, but could be used during post-processing to reduce the number of time steps realized during ABM-based analyses, increasing simulation speed. As $\xi$ approaches the study period, the `Upperbound` downsampling results in a more homogeneously weighted random mixing graph of contacts, resembling compartmental models with less heterogeneous preferential mixing among compartments. `Upperbound` maintains the density of the contact graph during downsampling.

While the investigation of the effects of `Upperbound` was motivated predominantly by its theoretical properties, it bears noting that some technologies—such as privacy-preserving or battery-sensitive contact tracking and reporting systems—do perform similar temporal aggregation of contact information over a period of time [56]. `Snapshot` performs temporal quantization in a sampling context. `Upperbound` performs both temporal quantization and aggregation via accumulation across that interval.

## SEIR simulation

We built an agent-based SEIR transmission model (`SEIR-ABM`) with the proximate contacts derived from synthetic proximity contact data collected with different observation frequencies. These synthetic proximity contact data are generated by downsampling with both `Snapshot` and `Upperbound` methods across datasets and diseases/pathogens.

**Agent-based SEIR model.** The agent-based model treats each person in the study population as an actor with one of four possible states with respect to a natural history of infection: *Susceptible*, latently infected (*Exposed*), *Infective*, and in a *Removed* state conferring persistent immunity to future infection. At any one time quantum, a given agent is further parameterized by a vector of active contacts, as specified by the proximity contact data for that agent for the current study, at the current level and type of aggregation.

Our SEIR agent-based model (`SEIR-ABM`) takes proximity contact data $\mathcal{D} = (\{G_{t_i}\}, \xi), \ t_i \in [i\xi, (i+1)\xi), \ i = 0, 1, \cdots, n-1$, an initial infected agent $\mathcal{V} \in \bigcup_i V_{t_i}, \ i = 0, 1, \cdots, n-1$, and a disease/pathogen $\mathcal{M}$ from the set {flu, SARS, fifth, pertussis, measles, chickenpox, MERS, diphtheria, COVID-19 wild type, COVID-19 Alpha variant, COVID-19 Beta variant, COVID-19 Delta variant}. The proximity contact observations were repeated four times, ensuring at least four months of proximity contacts time series for transmission simulation, to avoid underestimation of attack rate induced by right-censored data—particularly considering diseases/pathogens whose *Exposed* and *Infectious* periods add up to more than twenty days.

For each disease/pathogen, we gathered the basic reproductive number $R_0$ (Table 1) and range estimates of the latent period and infectious periods. Descriptions and comments about these diseases/pathogens can be found in the S1 Appendix. For simplicity, we will refer to a pathogen or a variant type of a pathogen as a disease in the following text. Each agent in the `SEIR-ABM` was associated with a latent period and personal infectious period drawn uniformly from corresponding ranges. Although in practice $R_0$ varies along with the rate of human-human or human-vector interactions spatially and temporally [57], we assumed identical $R_0$ for scenarios with different SHED datasets, because participants spend a considerable amount of their time on campus. This paper focuses on analyzing the impact of temporal resolution of Bluetooth discovering sensed proximate contact. Even if the $R_0$ is not calibrated

**Table 1. Disease parameter table.**

| Disease Name | Basic Reproductive Number | Incubation Period | Infectious Period |
|---|---|---|---|
| chickenpox | [†]15 [58] | 10—12 [59] | [*]8—11 [60] |
| COVID-19 Wild Type | [†]2.5 [61] | 5.6—7.7 [61] | 3—7 [61] |
| COVID-19 Alpha Variant | [*]3.23 [62] | 5.6—7.7 [61] | 3—7 [61] |
| COVID-19 Beta Variant | [*]3.13 [62] | 5.6—7.7 [61] | 3—7 [61] |
| COVID-19 Delta Variant | [*]4.93 [62] | 5.6—7.7 [61] | 3—7 [61] |
| diphtheria | [†]6.5 [63] | [‡]2—5 [59] | [‡]14—28 [64] |
| fifth | 1.8 [65] | [*]6—11 [66] | [*]4—9 [66] |
| flu | [‡]1.31 [67] | 2.28—3.12 [67] | [‡]2.06—4.69 [67] |
| measles | [†]15 [63] | [*]5—10 [68] | [*]4—6 [69] |
| MERS | 0.69 [70] | 2—14 [71] | [*]1—5 [72] |
| pertussis | [†]14.5 [63] | 7—10 [59] | [‡]14—21 [59, 73] |
| SARS | 3.6 [74–76] | 2—10 [75] | 4—14 [77] |

We use ([†]) for parameters derived as midpoint of reported range, ([‡]) for parameters derived range from different reports, ([*]) for parameters derived from starting range plus average duration, and ([*]) for parameters derived from other disease or comparative estimations.

separately for the specific population represented in each SHED dataset, it will not block us from interpreting how $R_0$ changes with the temporal resolution.

A simplified model was employed because we were primarily interested in the impact of measurement frequency. That model supports a stylized notion of the characteristics of the diseases explored, under a variety of epidemiological contexts:

- **Closed-population** Despite the fact that some for the 12 communicable diseases examined here are potentially lethal, we assume a closed population with no mortality or care-seeking that would cause an infected individual to be removed from circulation prior to recovery.

- **No intervention** Occurrence of infection within an individual or public health messaging regarding an identified outbreak can lead to the adoption of personal protective behavior such as elevated hygienic adherence and social distancing by population members; outbreaks can also lead to triggering of public health interventions, such as outbreak response immunization campaigns, quarantine efforts, contact tracing or increased vaccination. Within our simulation, we assume that infection status does not change agent behavior.

- **Consistent stages of infection** While the different communicable diseases considered in this paper differ considerably in the features of their natural history of infection (*e.g.*, the presence of both symptomatic and alternative oligo-/asymptomatic pathways, lack of permanent immunity) and routes of transmission (*e.g.*, airborne, droplet, fecal-oral), to focus on the effects of temporal quantization, we treated them as all being characterized by a 4-stage natural history of infection and proximity-based transmission, and as differing merely in terms of a disease-specific residence time within each state. This structure proceeds from *Susceptible* to *Exposed*, *Infectious*, and *Removed* states. In light of the 4-month time horizon of the model, we assumed that no re-infection is possible for each of the 12 communicable diseases.

- **Homogeneous infectious rate** We assume that for every discordant pair of individuals engaged in contact, the probability that the pathogen will be transmitted is governed by a constant hazard rate and the duration of the contacting period. This hazard rate is determined by a rate of potentially infecting exposures $\beta$ (for example, sneezing, aerosol production or hand-shaking), and a transmission probability per such exposure.

A System Dynamics/compartmental SEIR model (`SEIR-SD`) typically has $R_0 = \lambda \cdot \gamma^{-1} = \beta \cdot \bar{c} \cdot \gamma^{-1}$, where $\lambda$ is the force of infection, $\beta$ is the probability of transmission per contact between a susceptible and an infective, $\bar{c}$ is the average number of contacts made by each susceptible per unit time, and $\gamma$ is the rate at which an infectious person recovers or otherwise transitions to the *Removed* state. In the `SEIR-ABM`, because of the no intervention assumption, we estimate $\bar{c} = \frac{1}{T\|V\|}\sum_{i,j,k} \mathbb{I}(e_{jk} \in E_{t_i})$ given observed temporal graphs

$\{G_{t_i} = (V_{t_i}, E_{t_i})\}, \ t_i \in [i\xi, (i+1)\xi), \ i = 0, 1, \cdots, n-1$, where $T = \xi n$ is the effective study period, $\mathbb{I}$ is the indicator function, $\|V\|$ is the number of participants whose contact networks are recorded and $\|V\|$ does not change within the model because of the closed-population assumption.

All agents in the `SEIR-ABM` start as susceptible, with the exception of one initial infective. To address the potential impact on an outbreak outcome of the index infective individual, we iterate the initially infected person over the entire population. For each initial infection setting, we simulated the model across 30 distinct realizations, each associated with a distinct random number seed. At a high level, the algorithm of our `SEIR-ABM` can be summarized as follows (Algorithm 1):

**Algorithm 1**: Outline of SEIR-ABM

```
input: contact data D,
       disease M,
       initially infected person V
output: list of infectious events R
1 (G,ξ) ← D;
2 it ← iterator(G);
3 Vs ← init_population;
4 set_health_state (V, Infectious);
5 t ← 0;
6 G ← next(it);
7 while t < T do
8    map(update_health_state, Vs);
9    map(λv. append(R, expose_all_connected(v, G)), filter(λv.
at_health_state(v, Infectious), Vs));
10   if get_timestamp(G) + ξ < t then
11     G ← next(it);
12   end
13   t ← tick_tock(t);
14 end
```

**Parameter variation grid.** For each of the two downsampling methods, our simulation considers scenarios involving all combinations over three parameter classes: underlying populations, diseases, and downsampling intervals (mimicking observation frequencies). This paper specifically investigates the impact of the downsampling method and downsampling intervals given a specific underlying study population and pathogen. We consider all combinations of the following:

- Two downsampling methods: `Snapshot` and `Upperbound`

- Five datasets (SHED1–2, SHED7–9) with populations {39, 32, 61, 74, 78}, considering each possible exogenously infected index within each population

- Twelve pathogens and their accompanying communicable diseases: influenza type A, SARS-CoV, parvovirus B19, *Bordetella pertussis*, *Measles morbillivirus* (MeV), varicella-zoster virus (VZV), MERS-CoV, *Corynebacterium diphtheriae*, SARS-CoV-2, SARS-CoV-2 (B.1.1.7), SARS-CoV-2 (B.351), SARS-CoV-2 (B.1.617.2)

- Seven sampling intervals: 5 minutes (baseline), and 6 downsampling intervals: 10, 30, 60, 90, 180, 360 minutes

- An ensemble of 30 Monte Carlo realizations per parameterization

Considering all combinations of the above, we simulated 1312080 realizations of the `SEIR-ABM` model. Realizations were evaluated on a server with an Intel Xeon CPU E5–2690 v2 and 503GB memory. Models were created in AnyLogic 8.1.0 and exported to a standalone Java application with OpenJDK 1.8.0_252, resulting in 85GB of output data.

## Impact metrics

The transmission dynamics that emerged from an SEIR agent-based model have various usages, typified by evaluating interventions [78, 79], understanding transmission paths [80, 81], estimating disease parameters [82, 83], and forewarning outbreaks [84, 85]. These usages often have their basis on model simulated results, such as attack rates, transmission pathways in contact networks, and individual infection risks. We employed corresponding metrics to summarize changes in these simulation results across the parameter variation grid, bearing

variations due to stochastics of Monte Carlo realizations and rotations of the index infective within each population.

**Cumulative cases.**　The cumulative cases of a realization are the number of endogenous infections throughout that realization, starting from an infectious due to exogenous infection (not counted) until no one at the states of *Exposed* or *Infectious*. Because of assumptions as to the closed-population and acyclic stages of infection and persistent immunity, the cumulative cases are capped at one less than the size of underlying population. Without imposing assumptions on the distribution of cumulative cases over thirty Monte Carlo realizations, we employed median and inter-quantile range (IQR) as statistics to summarize cumulative cases by groups. In an agent-based model, the results of disease spread can be strongly influenced by the contact network of the initially infected individual. We explored two approaches to grouping the cumulative cases by constructing blocks with/without the index infectives.

**Outbreaks and outbreak-timing.**　Outbreak timing and behavior are commonly studied characteristics of communicable diseases, yet the quantifiable definition of an outbreak varies due to challenges regarding data collection and characterization of the appropriate cohort to be counted. Instead of imposing a quantitative definition, this work employs cumulative cases over time as a quantification of outbreak dynamics of disease in simulations for a given underlying population $V$ and observed contact data $\mathcal{D}$.

Each realization of the ABM-SEIR simulates an observation of cumulative cases over time as a continuous time series $\zeta^{(i)}(t; \mathcal{V}, \mathcal{D}_{\xi,\eta}, \mathcal{M})$, where $i = 1, 2, \cdots, N_{\mathrm{MC}}$ is the index of the Monte Carlo replication and $t$ is the time within the simulation, ranging from zero up to the time of termination of that realization. Recall that a realization terminates once its last infectious individual has finished their infectious period, thereby becoming *Removed*. The continuous time series of cumulative cases varies under the circumstances specified by the initial infectious individual $\mathcal{V}$ (specific to the underlying population $V$), observed contact data $\mathcal{D}_{\xi,\eta}$ (specific to the sampling method $\eta$ and sampling interval $\xi$), and disease $\mathcal{M}$ (which parameterizes the latent period and infectious period). The average of $\zeta^{(i)}$ over different initial infectious individual $\mathcal{V}$ and the Monte Carlo replication index $i$, denoted by $\hat{\mu}_{\zeta}(t; \mathcal{D}_{\xi,\eta}, \mathcal{M}) = \frac{1}{N_{\mathrm{MC}}\|V\|} \sum_{\mathcal{V} \in V, \ i=1,2,\cdots,N_{\mathrm{MC}}} \zeta^{(i)}(t; \mathcal{V}, \mathcal{D}_{\xi,\eta}, \mathcal{M})$, is the sample mean of cumulative cases given the underlying population $V$ and observed contact data $\mathcal{D}_{\xi,\eta}$ for the disease $\mathcal{M}$. Assuming a homogeneous chance for each individual of the underlying population $V$ to become the initial infectious individual, we can use $\hat{\mu}_{\zeta}(t; \mathcal{D}_{\xi,\eta}, \mathcal{M}) \in [0, \|V\|]$ to estimate the expected cumulative cases.

To compare across underlying populations of differing population sizes $\|V\|$, we defined the normalized expected cumulative cases (NECC) as $\hat{\varrho}(t; \mathcal{D}_{\xi,\eta}, \mathcal{M}) = \frac{1}{\|V\|} \hat{\mu}_{\zeta}(t; \mathcal{D}_{\xi,\eta}, \mathcal{M})$, where $\hat{\varrho}(t; \mathcal{D}_{\xi,\eta}, \mathcal{M}) \in [0, 1]$, with 0 indicating no infection and 1 indicating that the entire population is infected by time $t$. For a disease $\mathcal{M}$, given the underlying population $V$ and observed contact data $\mathcal{D}_{\xi,\eta}$, the NECC reflects the estimated expected fraction of maximum potential cumulative cases at time $t$.

**Attack rates.**　The attack rate of a realization is the ratio of cumulative cases to the size of its underlying population. The attack rate reflects the proportion of people who become infected started with an exogenously infected index in an otherwise susceptible population under our assumptions. Considering the attack rate as corresponding cumulative cases normalized by the size of its underlying population, we can compare attack rates among different underlying populations for a given disease, but with different downsampling methods and observation frequencies. These comparisons may shed light on whether an underlying

population will alter the importance of the temporal resolution to transmission simulation results of our interests.

We introduced an accuracy-precision view to measure the deviation with respect to the attack rate of simulations parameterized by the downsampled contact observations $\mathcal{D}_{\xi_+}$, $\xi_+ \in \{10, 30, 60, 90, 180, 360\}$ from the baseline $\mathcal{D}_{\xi_0}$, $\xi_0 = 5$. This deviation measurement is conducted for each downsampling method (`Snapshot` or `Upperbound`) under the circumstances parameterized by underlying population $V$ and disease $\mathcal{M}$. An underlying population $V$ shapes both the baseline contact observation $\mathcal{D}_{\xi_0}$ and the closure of the set of possible initial infectious individuals $\mathcal{V}$. We summarized two statistics: median and IQR, across the values of the attack rate drawn from an ensemble 482 of 30 realizations for each scenario defined by observations of contact network $\mathcal{D}_{\xi_+, \eta}$, initial infectious individual $\mathcal{V}$, and a type of communicable disease $\mathcal{M}$. The differences of a statistic (either the median or IQR) with respect to the attack rate of simulations parameterized by a downsampled contact observations $\mathcal{D}_{\xi_+}$ from the baseline $\mathcal{D}_{\xi_0}$ is defined as $\Delta \text{statistic}_{\xi_+} = \text{statistic}_{\xi_+} - \text{statistic}_{\xi_0}$.

**Transmission pathways in contact networks.**   Studies on transmission pathways in contact networks investigate how a disease/pathogen may spread on routes of transmission available between infectious and susceptible individuals, given the structure of contact networks where they reside [86, 87]. Sensor-data-derived proximate contacts reveal contact networks to study transmission pathways for diseases relying on routes of aerosol transmission and potentially direct contact transmission [29, 31, 88]. An infection pair of a realization, denoted by an ordered tuple of ($V_{\text{susceptible}}$, $V_{\text{infectious}}$), states the infection of $V_{\text{susceptible}}$ by $V_{\text{infectious}}$ during the infectious period of $V_{\text{infectious}}$ in the realization. Because infection pairs are elemental results reflecting transmission pathways from a realization, we sought to measure impacts of temporal resolution on transmission pathways in contact networks by comparing statistics of infections pairs from corresponding realizations.

Given a set of realizations from the `ABM-SEIR` model with a size $\|V\|$ population, if we put all possible tuples of individuals $\mathcal{T} = \{(i, j) \mid i, j \in V \wedge i \neq j\}$ into a canonical form with a rule $(i, j) \prec (k, l) \Leftrightarrow i \prec j \vee (i = k \wedge j \prec l)$, then we can express the frequencies of infection pairs in the set of realizations as a vector $\mathbf{\Omega} \in \mathbb{N}_0^{\|\mathcal{T}\| \times 1}$, where $\|\mathcal{T}\| = \|V\|^2 - \|V\|$. Assuming $\mathbf{\Omega} \neq 0$ and an uniform prior of an individual becoming the exogenously infected index, the $L_1$-normalized vector of infectious pairs' frequencies, denoted by $\frac{\mathbf{\Omega}}{\|\mathbf{\Omega}\|_1}$, is the relative risk of infection pairs occur in a realization.

Now we consider two parameter sets $\mathcal{P}_p$ and $\mathcal{P}_q$ sharing a disease/pathogen, an underlying population, and a sampling method, but with different duty cycle intervals $\xi_p$ and $\xi_q$. The differences of realizations resulted by $\mathcal{P}_q$ from $\mathcal{P}_p$, in terms of frequencies of infection pairs, can be reflected by a weighted-Minkowski distance of order one, denoted by

$$D_{\mathrm{M}}\left(\mathbf{\Omega}_p, \mathbf{\Omega}_q\right) = \left(\frac{\mathbf{\Omega}_p}{\|\mathbf{\Omega}_p\|_1}\right)^{\top} \cdot \left(\mathbf{\Omega}_p - \mathbf{\Omega}_q\right)^{\text{abs}},$$

where $\frac{\mathbf{\Omega}}{\|\mathbf{\Omega}\|_1}$ is the weight and $(\cdot)^{\text{abs}}$ is the element-wise absolute value operator for a vector, such that $\mathbf{\Omega}^{\text{abs}} = [\text{abs}(\Omega_1) \cdots \text{abs}(\Omega_{\|\mathcal{T}\|})]^{\top}$. When $\xi_p < \xi_q$, $\mathcal{P}_q$ is a parameter set with a larger duty cycle interval than $\mathcal{P}_q$, this weighted-Minkowski distance between frequencies of infection pairs $\mathbf{\Omega}_p$ and $\mathbf{\Omega}_q$ can be interpreted as the risk-weighted $L_1$-distance resulted by a downsampled proximate data of duty cycle interval $\xi_q$ from a reference proximate data of duty cycle interval $\xi_p$. This expected $L_1$-distance handles variations due to stochastics of Monte Carlo realizations and rotations of the index infective within each population, allowing us to infer impacts of observation frequencies on simulated results of infection pairs sharing an underlying population. Notice that $0 \leq D_{\mathrm{M}}(\mathbf{\Omega}_p, \mathbf{\Omega}_q) \leq N_{\mathrm{MC}} \|V\|$,

where $N_{\mathrm{MC}} = 30$ is the number of Monte Carlo realizations per parameterization, and $\|V\|$ is due to the rotation of index infectives. We have $D_{\mathrm{M}}(\boldsymbol{\Omega}_p, \boldsymbol{\Omega}_q) = 0$ when $\boldsymbol{\Omega}_p = \boldsymbol{\Omega}_q$, and $D_{\mathrm{M}}(\boldsymbol{\Omega}_p, \boldsymbol{\Omega}_q) = N_{\mathrm{MC}}\|V\|$ when $\nexists \; i, j \in V, \; \boldsymbol{\Omega}_p^{(i,j)} > 0 \land \boldsymbol{\Omega}_q^{(i,j)} > 0 \land i \neq j$. To unify the scale of $D_{\mathrm{M}}(\boldsymbol{\Omega}_p, \boldsymbol{\Omega}_q)$ for underlying populations with different sizes, we employed $D_{\mathrm{NM}}\left(\boldsymbol{\Omega}_p, \boldsymbol{\Omega}_q\right) = \frac{D_{\mathrm{M}}(\boldsymbol{\Omega}_p, \boldsymbol{\Omega}_q)}{\|V\|}$ and $0 \leq D_{\mathrm{NM}}(\boldsymbol{\Omega}_p, \boldsymbol{\Omega}_q) \leq N_{\mathrm{MC}}$.

**Individual infection risks.** The infection risk of an individual is estimated by the fraction of realizations where the individual got infected. For a population $V$, its individual infection risks under a circumstance of a disease, a sampling method, and an sample interval can be presented by a vector of infection risks for everyone in the population, denoted by $\boldsymbol{\Psi} \in \mathbb{N}_0^{\|V\|}$. The $L_1$-normalized vector of individual infection risks, denoted by $\boldsymbol{\rho} = \frac{\boldsymbol{\Psi}}{\|\boldsymbol{\Psi}\|_1}$ can be considered as the likelihood of an individual to be the most likely infected.

For each individual given each combination of disease and datasets, we calculated the Laplacian-smoothed individual infection probability based on infection counts from simulations informed by $\xi_+$-downsampled contact data using downsampling method $\eta \in \{\text{Upper-bound, Snapshot}\}$; then we assembled the individual infection probability into a vector of the $L_1$-normalized vector of individual infection risks $\boldsymbol{\rho}(\mathcal{M}, \mathcal{D}_{\xi,\eta})$. Laplacian-smoothing was employed to ensure that those who were not infected in simulation outcomes are still assigned a small probability of being infected.

To characterize the deviation of $\boldsymbol{\rho}(\mathcal{M}, \mathcal{D}_{\xi,\eta})$ as sample interval $\xi$ increases, we employed Kullback-Leibler divergence of the baseline $\boldsymbol{\rho}(\mathcal{M}, \mathcal{D}_{\xi_0,\eta})$ from $\boldsymbol{\rho}(\mathcal{M}, \mathcal{D}_{\xi_+,\eta})$ with sample interval $\xi_+$, denoted as $\delta_{\xi_+} = D_{\mathrm{KL}}(\boldsymbol{\rho}(\mathcal{M}, \mathcal{D}_{\xi_0,\eta}) \| \boldsymbol{\rho}(\mathcal{M}, \mathcal{D}_{\xi_+,\eta}))$, where $\xi_0 = 5$, and $\xi_+ \in \{10, 30, 60, 90, 180, 360\}$. The KL-divergence $\delta_{\xi_+}$ is evaluated six times (at $\xi_+ = 10, 30, 60, 90, 180, 360$) under circumstances parameterized by combinations of the downsampling method $\eta$, underlying population $V$, and disease $\mathcal{M}$.

## Results

We evaluated the impact of downsampling methods and frequencies from two perspectives: the resultant distortions of network structure, and deviation in transmission model outcomes. Each such evaluation employed the results of the baseline fidelity network representation as the reference for assessing such distortions/deviations. The network structure analyses show that, as observation frequency reduces, the `Snapshot` method and the `Upperbound` method distort network structure in different ways—the `Snapshot` keeps the average cumulative contact time at the cost of underestimating node degrees. In contrast, the `Upperbound` method results in inflated average cumulative contact time but retains the node degree distribution. The evaluated the deviation in transmission model outcomes at both the population and individual levels are analyzed in the following sections. To keep main text succinct, we restricted figures in this section to three selected diseases—influenza type A, denoted as `flu`; SARS-CoV-2 (B.1.617.2), denoted as `covid19delta`; and *Measles morbillivirus* (MeV), denoted as `measles`. Corresponding figures that include all 12 diseases can be found in the supplementary materials.

### Impacts on population-level simulation results

The impacts of the observation frequency on simulation results from the `ABM-SEIR` model can be considered at the population and/or individual level. Cumulative cases and attack rates were used to measure the impacts of observation frequency on simulation results at the population level. Such population-level results of a transmission model are often used to evaluate the size of the outbreak or the overall severity of an upcoming wave. We performed Welch's

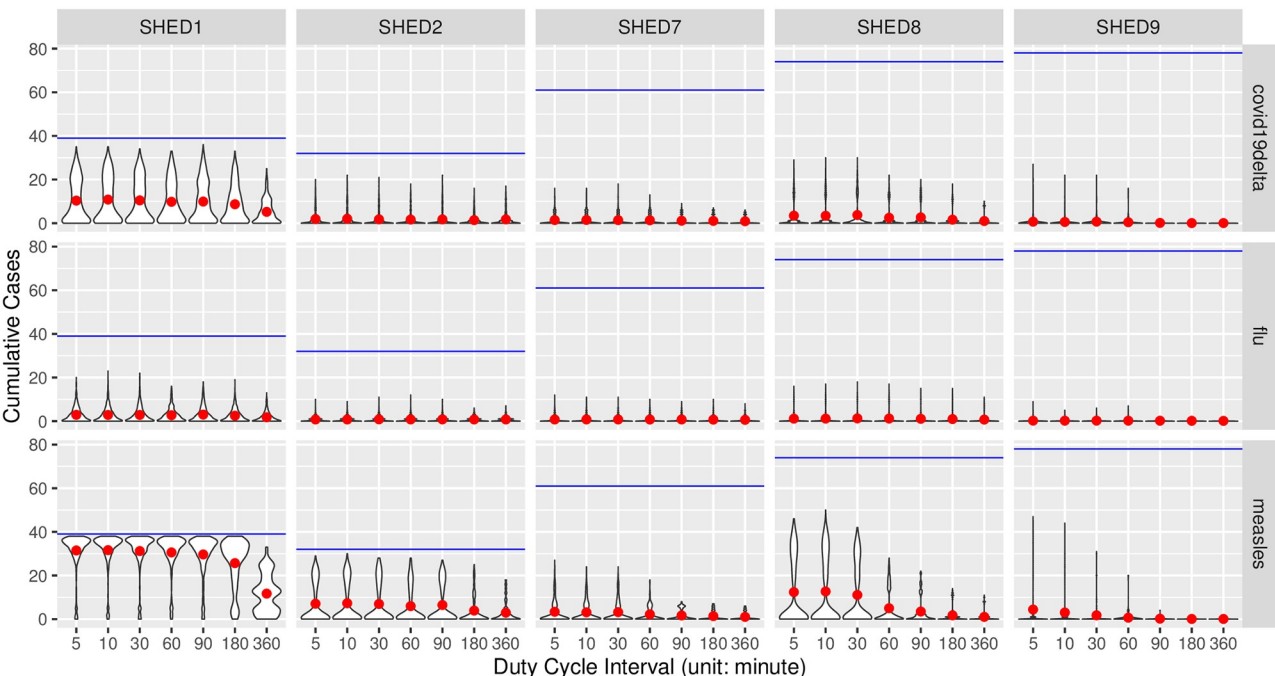

**Fig 1. Grids of violin plots of cumulative cases for the `Snapshot` method.**

$t$-test on cumulative cases with different $\xi$ to draw quantitative conclusions as to the impact of observation frequency on the mean of cumulative cases. We used the Prentice modified Friedman test on cumulative cases with different $\xi$ to test the impact of observation frequency on the distribution of cumulative cases.

**Cumulative cases.**   In Figs 1 and 2, we visualized the empirical distributions of the cumulative cases in realizations of the agent-based SEIR model informed by downsampled contact data at different duty cycle intervals, with one grid for each of the `Snapshot` and the `Upperbound` downsampling methods. In general cases, the `Snapshot` method preserves the distribution of cumulative cases regardless of increasing duty cycle interval. Meanwhile, the `Upperbound` method suffers from systematically overestimating the plausibility of having an outbreak. For diseases having relatively high $R_0$ (*e.g.*, measles) and "closer" population (`SHED1-2`), the `Snapshot` method risks underestimating plausible outbreaks with sparse observations—those sampled an hour or more apart—while the `Upperbound` method retains the risk of corresponding outbreak occurrence at the cost of results varying between either universal infection or no further infection after the initial infection.

**Welch's $t$-test.**   We validated our interpretation of Figs 1 and 2 with the Bonferroni-corrected Welch's $t$-test [89, 90], provided by R package *stats*, version 4.0.2. For each sampling method of the `Snapshot` and `Upperbound`, we tested cumulative cases with different duty cycle intervals blocked by diseases and underlying populations. Resulting 60 blocks, with each group (observation frequencies) having at least 960 samples (cumulative cases of realizations), sufficiently large to consider the robustness of $t$-test given the distribution of cumulative cases' departure from normality [91, 92], as shown in Figs 1 and 2. Setting the alpha-value as 5%, our null hypothesis is that given a disease/pathogen other than high $R_0$ diseases (chickenpox, measles, pertussis) and a underlying population, the mean of cumulative cases resulted by proximate contacts collected with different observation frequencies are equal. For each block,

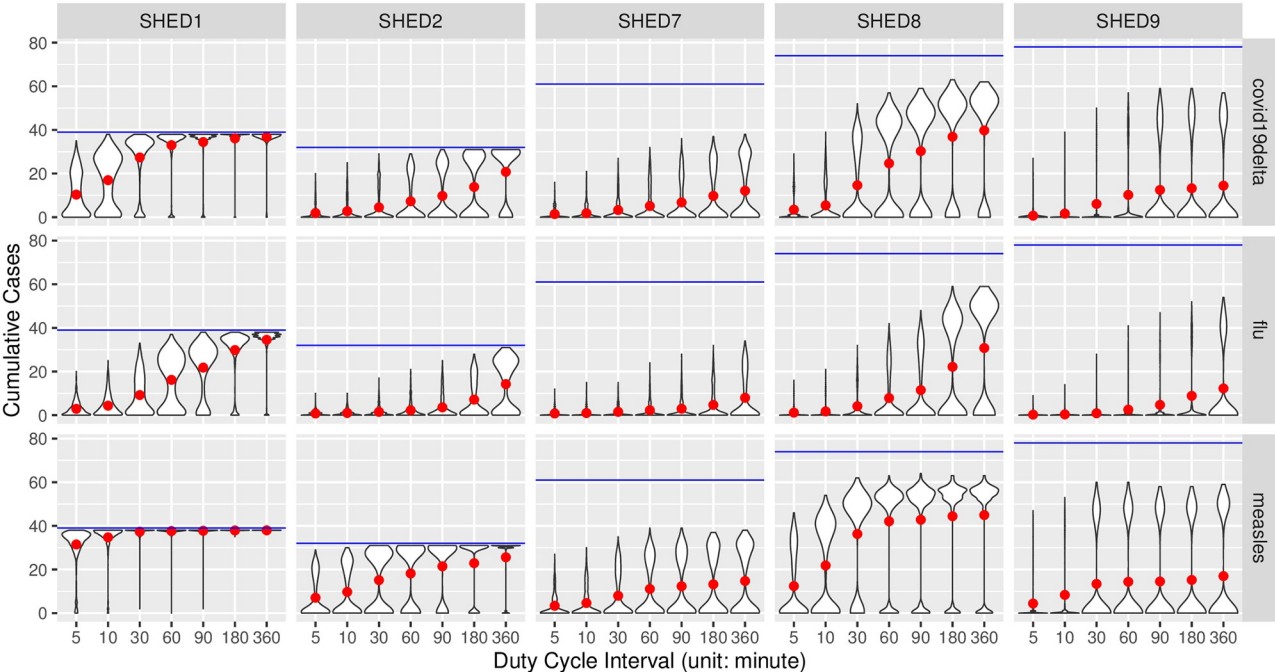

**Fig 2. Grids of violin plots of cumulative cases for the Upperbound method.** Figs 1 and 2 are two grids of violin-plots visualize the empirical distributions of the cumulative cases outcomes from the agent-based SEIR model informed by proximity contact data downsampled at different inter-observation intervals, given different diseases, within different underlying populations. At the highest level, we summarize simulated cumulative cases by downsampling methods, resulting in two subfigures having a similar arrangement, referred to here as a violin-plot grid. One such grid is present for each of the Snapshot and the Upperbound downsampling methods. Each such grid characterizes how cumulative cases vary by disease (row), and dataset (column). Within each cell of the violin-plot grid, violin-plots of cumulative cases are arranged from left to right according to the increasing inter-observation interval, with a 5-minute interval being the left-most, and 360-minute being the right-most. Each violin-plot shows the distribution of simulated cumulative cases over different random seeds and initially infected individuals at the inter-observation interval denoted by its horizontal axis label. The size of underlying population is denoted with blue lines, and the red dots indicate mean values of corresponding violin plot-denoted cumulative cases distributions.

pairwise by duty cycle intervals resulting 21 comparisons per block and $\alpha_{\text{altered}}$ = 0.05/21 = 0.00238. It turned out for the Snapshot method null hypotheses were not rejected for pairs of observation intervals less than or equal to 30 minutes, except for SHED8-diphtheria between pairs of duty cycle intervals 10–30 ($t$(4368.3) = −3.10115, $p$ = 0.00194), SHED8-SARS 5–30 ($t$(4294.5) = −3.23677, $p$ = 0.00122)), and SHED9-diphtheria 5–30 ($t$(4399.6) = −3.61278, $p$ = 0.00031). For the Upperbound method hypotheses are rejected, except for SHED2-fifth 5–10 ($t$(1910.7) = −1.8350, $p$ = 0.06666), SHED2-MERS 5–10 ($t$(1902.2) = −1.037, $p$ = 0.29989); SHED9-COVID19Delta 90–180 ($t$(4667.3) = −1.2404, $p$ = 0.21490), 180–360 ($t$(4675.8) = −1.9465, $p$ = 0.05165). Full results of Welch's $t$-tests can be found in S1 Table.

**Prentice-modified Friedman tests.**   We further validated our interpretation of Figs 1 and 2 with the Prentice modified Friedman test [93–95], provided by R package *muStat*, version 1.7.0. We tested cumulative cases grouped by sampling interval and blocked by data collection, sampling method, population (dataset), disease, and initial infection node. Resulting $\chi^2$ = 222081, with 6 degrees of freedom (reflecting the fact that the sampling interval $\xi \in \{5, 10, 30, 60, 90, 180, 360\}$ has 7 choices in total), and $p < 2.2e{-}16$, with the null hypothesis being that the sampling interval does not differentiate the distribution of cumulative cases, for the same data collection, sampling method, dataset, disease, and initial infection node.

**Outbreaks and outbreak timing.**   For each disease, we drew a grid of subplots to visualize the change of normalized expected cumulative cases (NECC) under the impact of the sampling

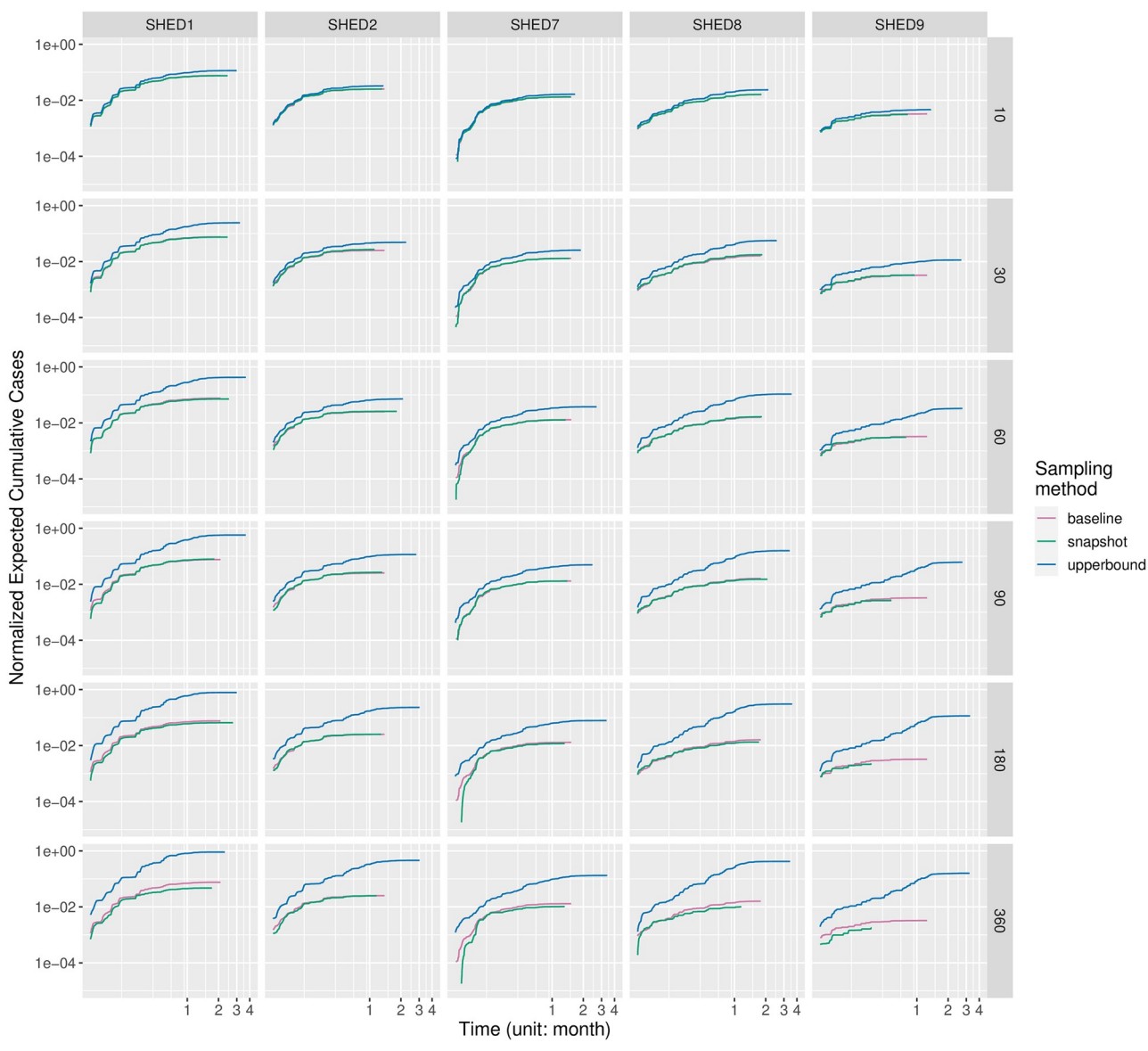

**Fig 3. Outbreak timing of flu.**

interval (row) and sampling method (color) for each underlying population (column). Four representatives are selected and shown in Figs 3–6; the remainder can be found in the S1 Appendix. Similar diseases appear to have similar dynamics, as shown, for example, in Figs 3 and 4; diseases with extremely low and high $R_0$ tend to behave quite differently regardless of sampling method, interval, and dataset, as can be seen by contrasting Figs 5 and 6. In general, the NECC curves of the `Snapshot` method exhibit smaller estimates of cumulative cases (on the vertical axis) over time. As the downsampling interval increases, the NECC curves of the `Snapshot` method also tend to end earlier on the horizontal axis. This early termination reflects the fact that disconnections halt the pathogen spread among infectious and susceptible due to missed contacts. For a given underlying population, the early termination of disease

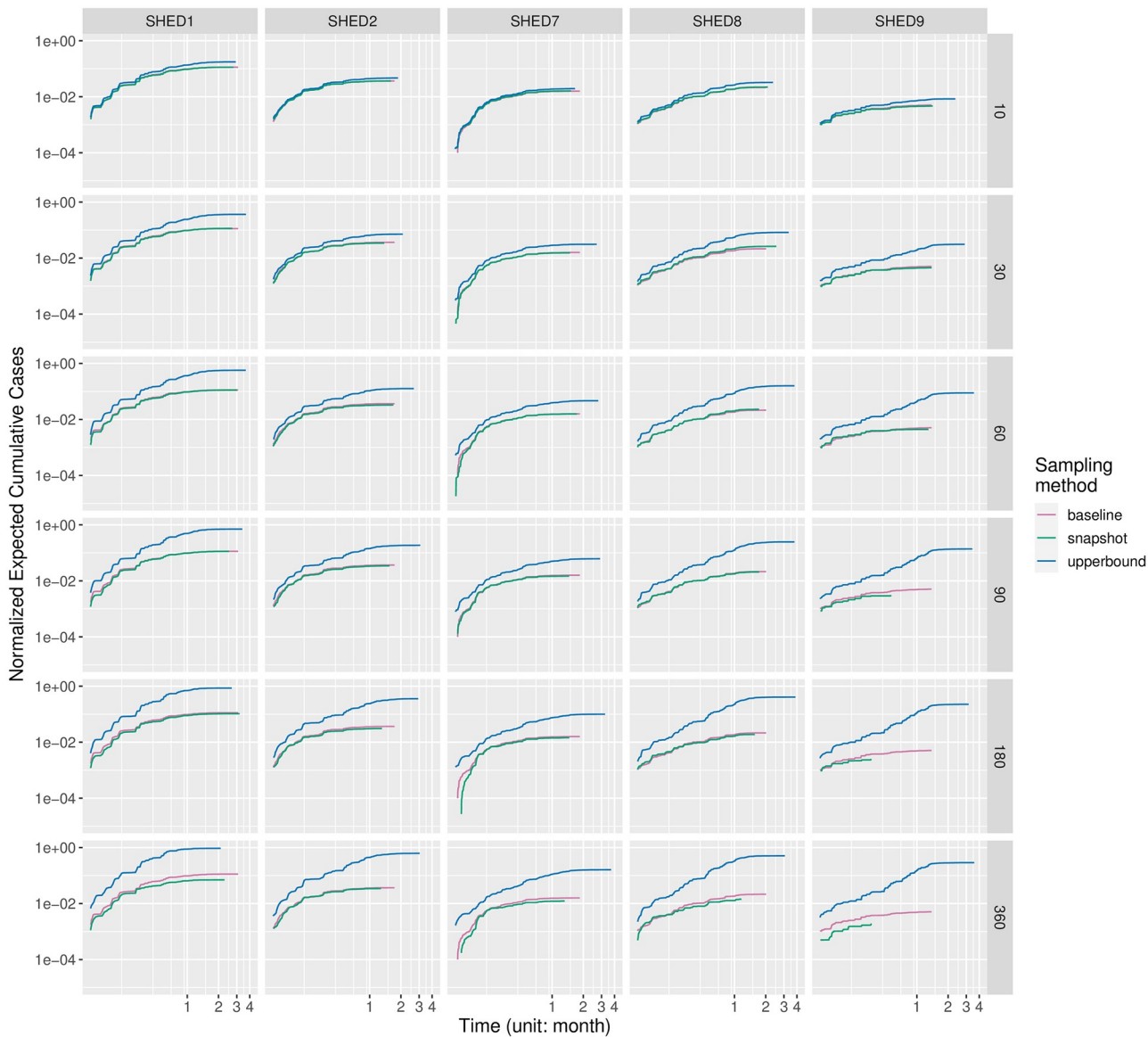

**Fig 4. Outbreak timing of SARS.**

spread seems consistent with the smaller estimates of cumulative cases, except for the scenarios when NECC curves terminated early due to the entire population having been infected. This analysis demonstrates that:

- As would be expected, given an initially susceptible population, a pathogen with a tendency to catalyze an outbreak will often exhibit apparent, sharp increases in infections during the outbreak period. Weakly spreading pathogens have an initial ascent followed by a long tail. More notable is that this tendency holds largely invariant of sampling method, population, and sampling interval.

- In a pattern that is maintained—*mutatis mutandis*—across populations, sampling method, and interval, clinically similar diseases exhibit similar curves: SARS (Fig 4) is known to have

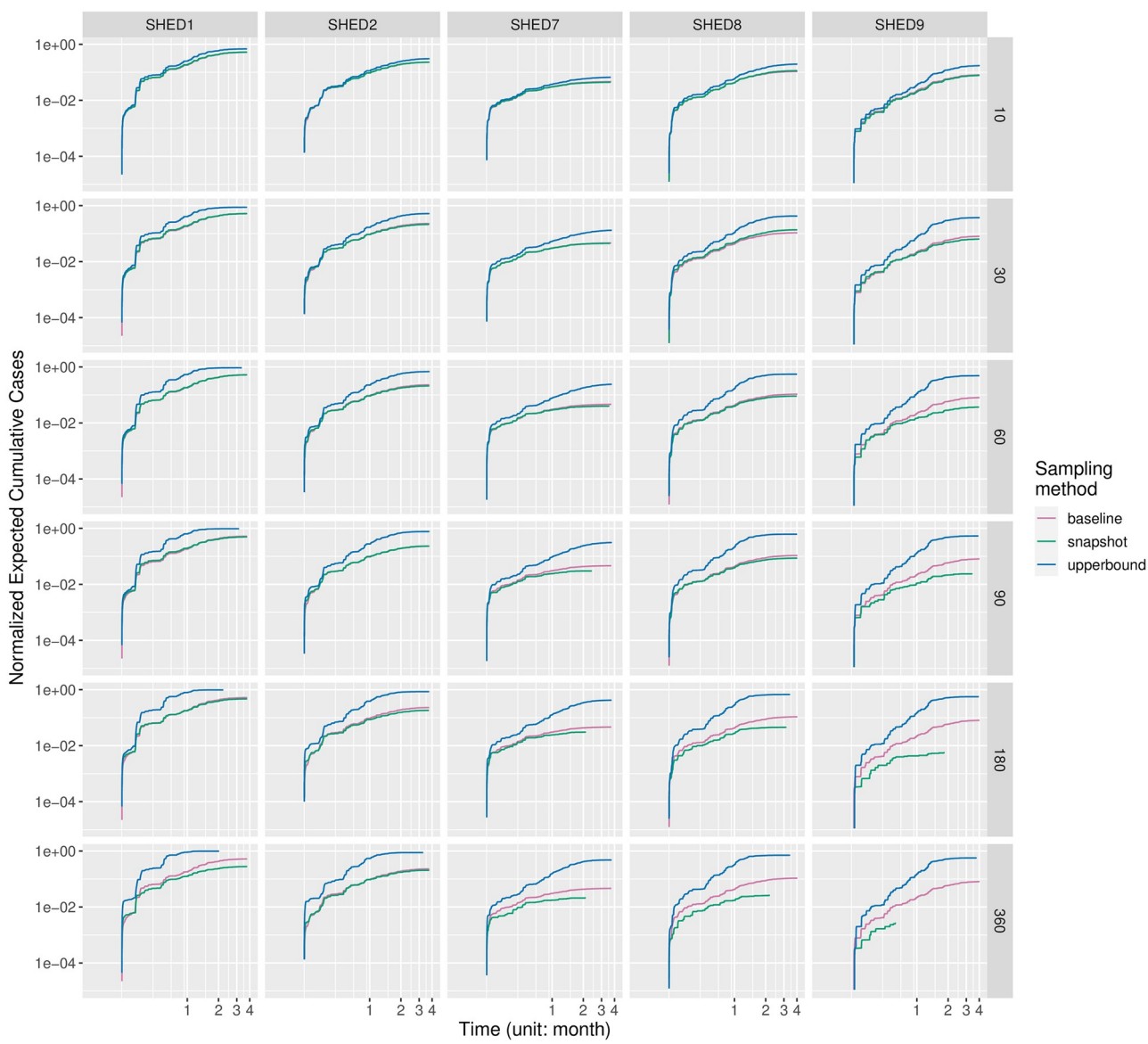

**Fig 5. Outbreak timing of pertussis.**

similar characteristics to flu (Fig 3), and they exhibit similar patterns for our measure. The discrepancy is small for "closer" populations.

- In a pattern that again holds independent of sampling method and interval as well as population (dataset), diseases with different $R_0$ behave differently. Pertussis (Fig 5) has the highest $R_0$ amongst the diseases we simulated, while MERS (Fig 6) has the lowest. Their pattern is distinct—pertussis tends to have a clearer outbreak. By contrast, SARS exhibits a steep curve in the beginning and a long tail, indicating limited disease spread.

- Discrepancies between Snapshot and Upperbound from the 5-minute baseline increased with the sampling interval $\xi$. Discrepancies induced by the sampling interval exert less impact than the characteristics of the study population, with "diffuse" communities (like SHED9) exhibiting substantial discrepancies for both Snapshot and Upperbound.

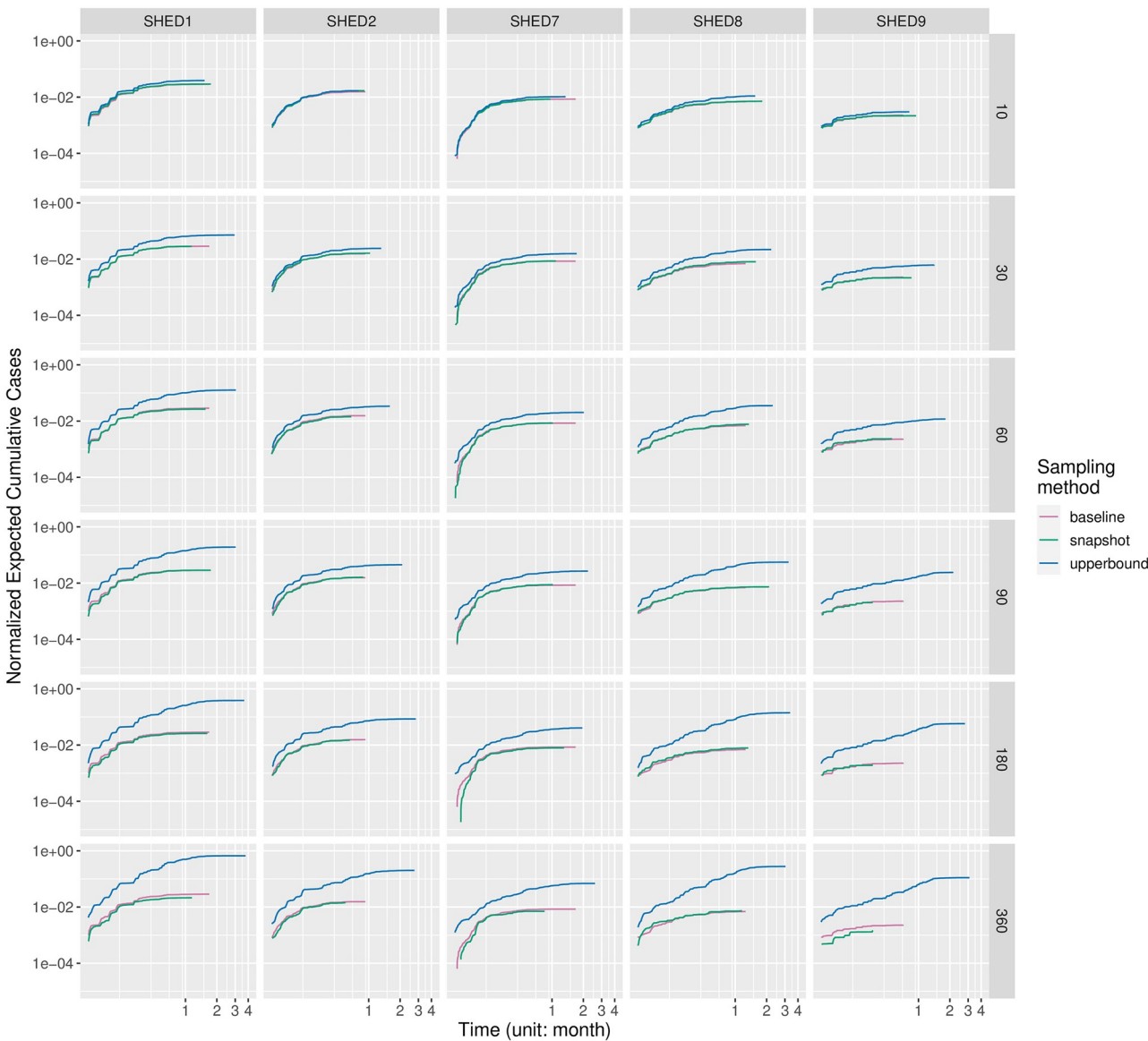

**Fig 6. Outbreak timing of MERS.** Figs 3–6 compares outbreak timing of diseases. The normalized expected cumulative cases (NECC) estimates the expected cumulative cases as a fraction of its underlying population size at time $t$, given the disease, underlying population, sampling method, and sampling interval. We organized NECC over time as colored curves in a grid of cells for each disease, with the underlying population in columns and sampling interval in rows. Each cell plots three NECC curves of the baseline (red), downsampled with the `Snapshot` method (green), and downsampled with the `Upperbound` method (blue). Each NECC curve plots NECC (y-axis) over time in the unit of months (x-axis). Both the x-axis and the y-axis are $\log_{10}$-scaled. The origin of each plot is at (0.5 weeks, $10^{-5}$). Minor ticks on the x-axis(lighter vertical strips) mark 1, 2, 3, and 4 weeks before the first month, then every half month (15 days). NECC curves have different periods (on the x-axis) because they either lack proximate contact from infectious to susceptible or run out of the susceptible. If a NECC curve ends early because of running out of susceptible, its NECC value will stop at one on the y-axis. Steeper NECC curves imply faster growth in cumulative cases. We selected four representatives here; the remainder can be found in the S1 Appendix.

- When the sampling interval is large, the `Snapshot` method outperforms `Upperbound` in terms of having NECC closer to the 5-minute baseline, at the cost of shortening the disease spreading period.

  **Attack rate.**   In the accuracy-precision view, as shown in Fig 7, when downsampling with the `Snapshot` method, points are closer to the origin for communicable diseases with low $R_0$

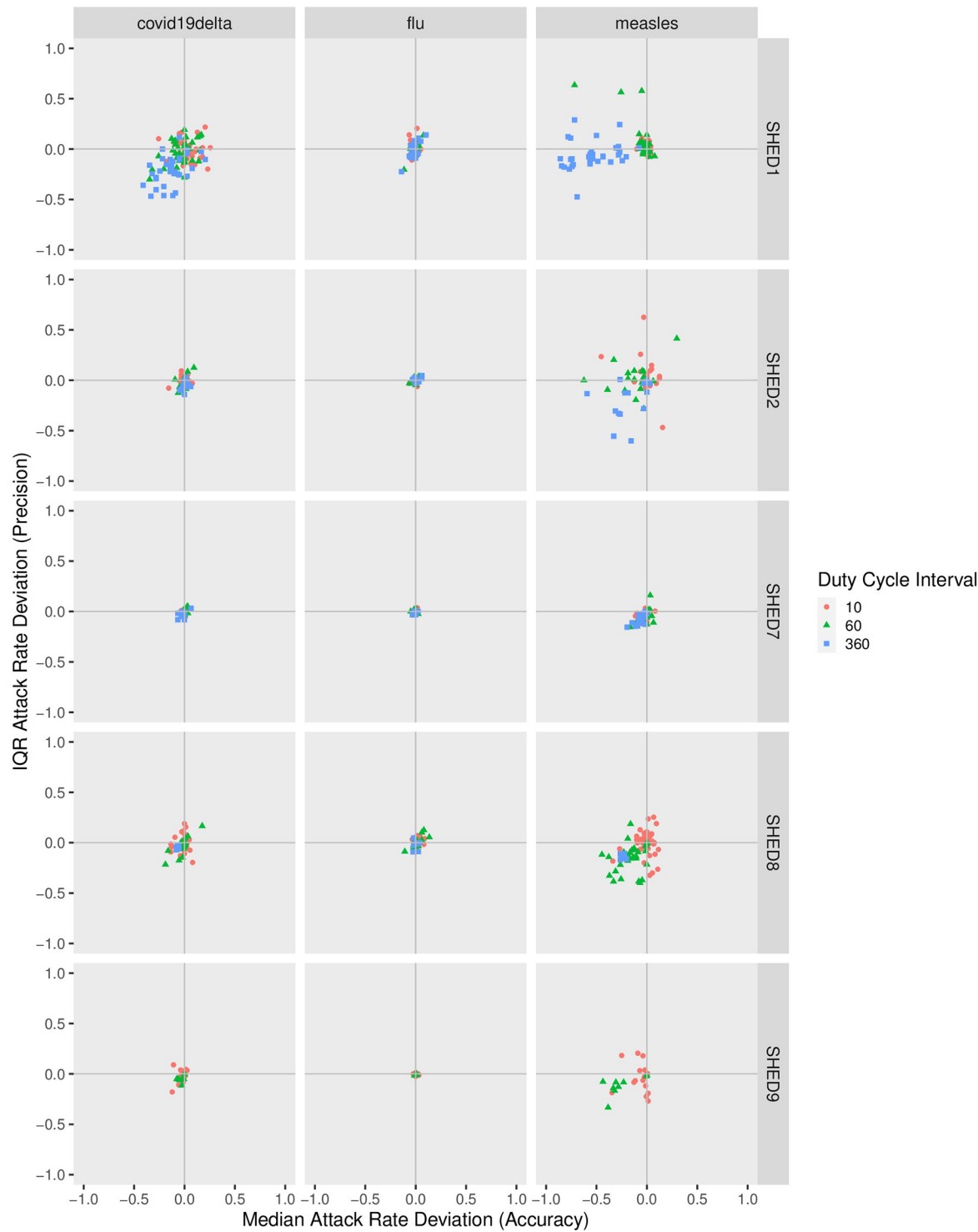

**Fig 7. Attack rate given initial infection node for the `Snapshot` method.**

and for "diffuse" population such as {SHED7, SHED8, SHED9}, indicating the advantage of `Snapshot` at maintaining an estimate of attack rate as downsampling interval increases. For diseases with high $R_0$ (measles) and "closer" communities {SHED1, SHED2}, `Snapshot` underestimates the attack rate as $\xi$ increases, whereas `Upperbound` overestimates.

`Upperbound` reduces IQR deviation of estimated attack rate while the `Snapshot` increases interquartile range (IQR) deviation.

The accuracy-precision deviation of simulation results in terms of attack rate depends on the underlying population structure ("closer" or "diffuse"), the type of communicable disease, the sampling method (`Snapshot` or `Upperbound`), and the sample interval. Casual inspection of the skewed nature of the distributions towards higher values of the horizontal axis within each subplot of Figs 7 and 8 (indicating increasing median deviation in incidence) confirms that increasing the sample interval results in over-estimation of the attack rate, as is suggested in [39]. By contrast, the clustering of the points by color in each subplot suggests that the initial infection node exerts a smaller impact on the two statistics we have chosen to reflect the accuracy-precision tradeoffs.

Comparing within subplots column-wise, the `Snapshot` method performs well with "diffuse" communities, resulting in both low deviation of median and low deviation of IQR. When used with "closer" networks, `Snapshot` tends to overestimate the attack rate but underestimate the IQR. `Upperbound` exhibits greater deviation than `Snapshot`, and is more consistent as sampling interval increases given other factors—from left to right. When sampling interval is brief and sampling rate high, attack rate exhibits low median and IQR deviation from the ground truth, because the reconstructed contact network is less distorted. As the sampling interval increases and sampling rate decreases, `Upperbound` tends to become both less accurate and less precise. As the sample interval increases further, the overestimation of the attack rate reaches a limit as people directly or indirectly connected to the initially infectious person are reliably infected for high $R_0$ pathogens, or people remain uninfected for pathogens with low $R_0$.

Comparing within subplots row-wise, disease-specific patterns are also visible: estimates for the attack rate of diseases with low $R_0$, such as influenza type A, seem relatively insensitive to the sampling interval. Pathogens/communicable diseases with sufficiently high $R_0$ tend to behave similarly as sampling interval increases, regardless of their differences in $R_0$ value.

## Impacts on individual-level simulation results

We measure the impacts of observation frequencies on the simulation results at the individual level with transmission pathways and individual infection risks. Individual risk of infection can suggest vulnerable group to prioritize resource allocation and ensure health equity [96]. Individual risk of infection is asymptotically approached by the fraction of realizations in which an individual is infected. The difference of individual risk of infection can be compared pairwise in terms of weighted-Minkowski distance among scenarios with different datasets $\mathcal{D}$ for the same underlying population $V$ and disease $\mathcal{M}$. We calculated the Kullback-Leibler divergence on individual infection probabilities with different $\xi$ to draw quantitative conclusions on the impact of downsampling frequency on simulation results at the population level. Higher KL-divergence values from the `Snapshot` method for `SHED9` were observed for chickenpox, COVID-19, diphtheria, measles, and pertussis, and are indicated by darker colors of the corresponding column on Fig 9. Lower KL-divergence values associated with influenza type A, regardless of dataset and downsampling frequency, induces its lighter color in the corresponding column in that figure. We find that the KL-divergence can effectively summarize the information shown on Fig 9 and therefore can serve as an efficient metric to measure differences in individual risk.

**Distance matrices of infection pairs.**   Fig 9 shows matrices of pairwise weighted-Minkowski distances of frequencies of infections pairs given downsampling methods, disease, and sampling frequencies for underlying populations, with the color shifting from lighter to darker

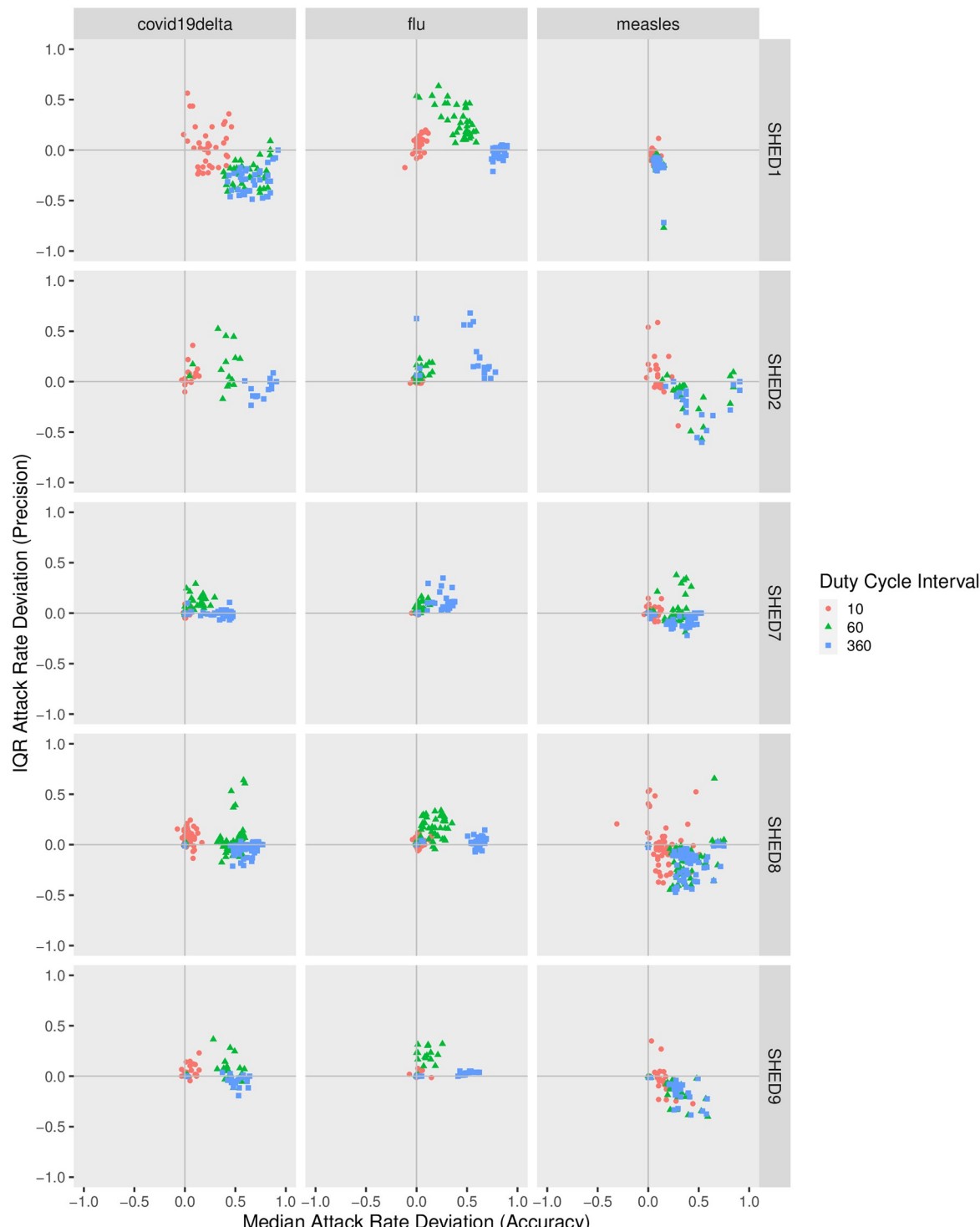

**Fig 8. Attack rate given initial infection node for the `Upperbound` method.** The accuracy-precision views in Figs 7 and 8 measure the deviation with respect to the attack ratio of simulations parameterized by the downsampled contact observations $\mathcal{D}_{\xi_+}$, $\xi_+ \in \{10, 30, 60, 90, 180, 360\}$ from the baseline $\mathcal{D}_{\xi_0}$, $\xi_0 = 5$. Here we only show $\xi_+ \in \{10, 60, 360\}$ to prevent points from being over-clustered, and the rest can be found in the appendix. Subplots are arranged as grids according to the combinations of underlying population $\mathcal{V}$ and disease $\mathcal{M}$. Within each subplot specific for a given combination of $\{\mathcal{D}_\xi, \mathcal{V}, \mathcal{M}\}$, deviation of the median attack rate is shown on the horizontal axis (reflecting accuracy) and deviation of the inter-quartile range (IQR) for attack rate is depicted on the vertical axis (negatively correlated with precision). Each datapoint within such a subplot is associated with a specific sample interval $\xi$ of $\mathcal{D}_\xi$,

whose value is denoted by both color and shape for visual clarity. In both Figs 7 and 8, points with same color and shape tend to cluster instead of mixing with other colors, indicating that sample interval impacts govern both the accuracy and precision of the attack ratio more than the initial infection node $\mathcal{V}$. In Fig 7, when downsampling with the `Snapshot` method, points are closer to the origin for diseases with low $R_0$ (except for chickenpox, measles and pertussis) and for "distant" population such as {SHED7, SHED8, SHED9}, indicating the advantage of `Snapshot` at maintaining an estimate of attack ratio as downsampling interval increases. For diseases with high $R_0$ (chickenpox, measles, pertussis) and "closer" populations ({SHED1, SHED2}, `Snapshot` metod underestimates the attack ratio as $\xi$ increases, whereas `Upperbound` slightly overestimates. `Upperbound` reduces IQR deviation of estimated attack ratio while the `Snapshot` increases IQR deviation.

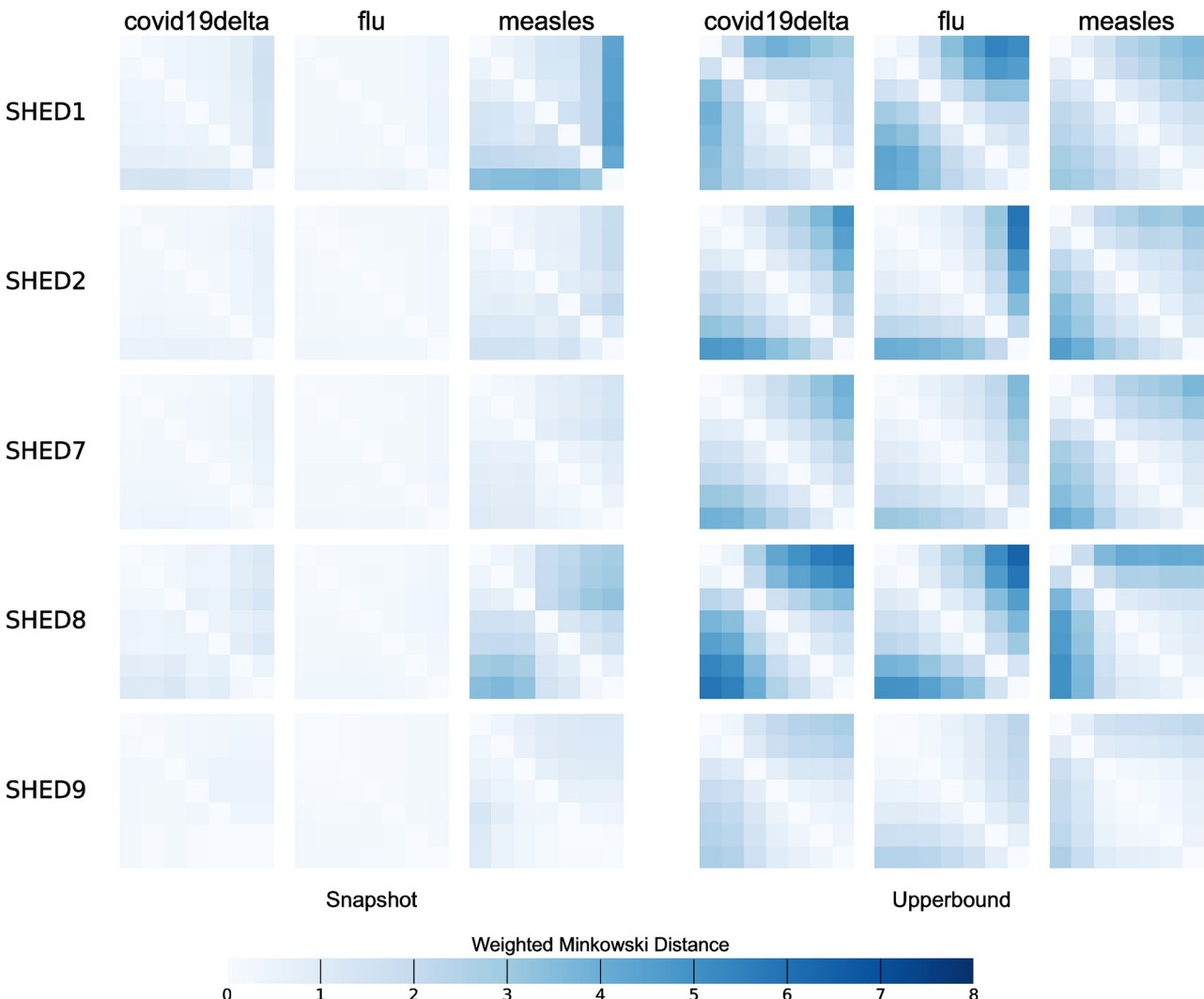

**Fig 9. Distance matrix of infection pairs.** Two grids of two dimensional (2D) histograms visualize matrices of pairwise weighted-Minkowski distances of frequencies of infections pairs given downsampling methods, disease, and sampling frequencies for underlying populations. All histograms share a color-scale as shown in the legend, with the color shifting from lighter to darker with the increasing degree of dissimilarity. For each matrix, starting from its top left corner, inter-observation intervals are arranged in ascending order—$\xi$ = 5, 10, 30, 60, 90, 180, 360—horizontally from left to right and vertically from top to bottom. We found `Snapshot` is better at preserving consistent frequencies of infection pairs, particularly with an observation frequency higher than once per half-hour, except for higher $R_0$ diseases in "diffuse" communities, such as chickenpox and measles in SHED9. For lower $R_0$ diseases (flu), particularly in "closer" communities like SHED1, the `Snapshot` method have weighted-Minkowski distance less than two even between the observation frequencies of 5-minute and 360-minute.

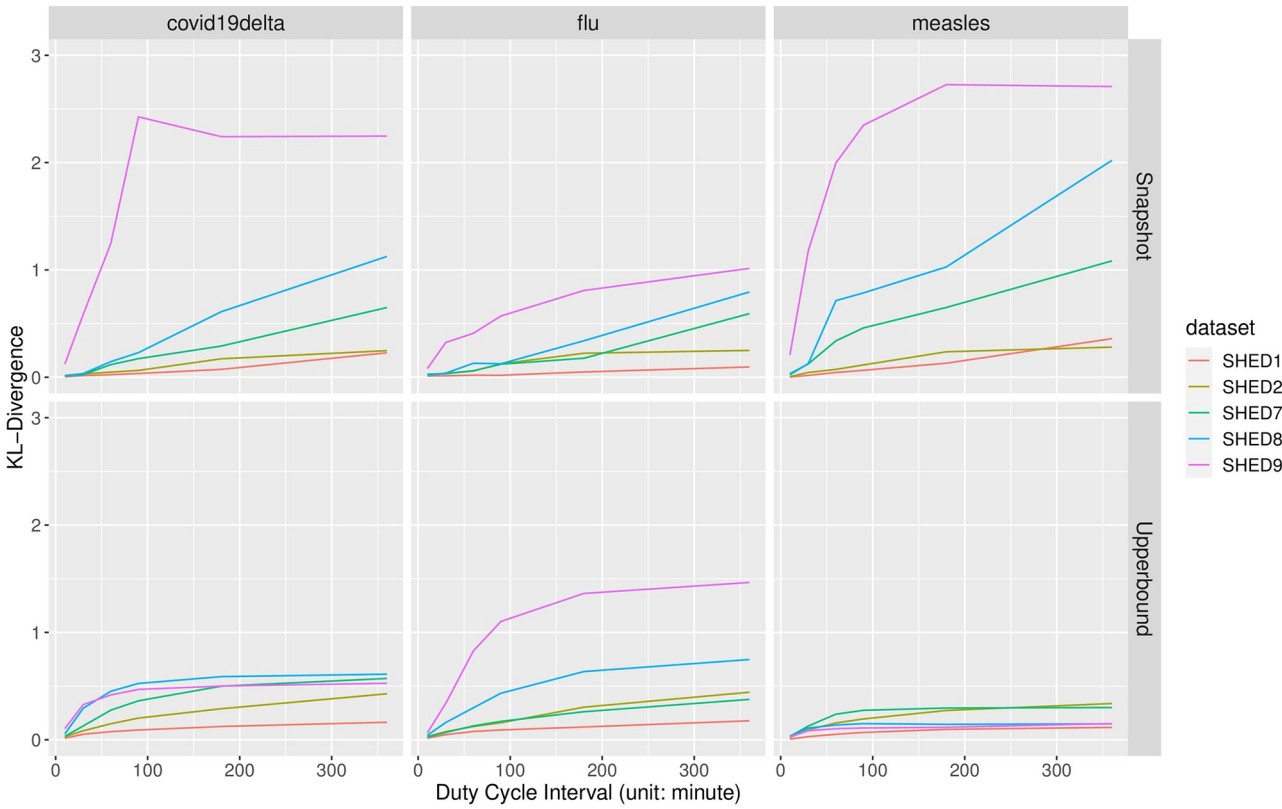

**Fig 10. Kullback-Leibler divergence of individual infection risk.** The deviation of individual infection risks as sample interval $\xi$ increases is visualized as a grid of line plots. Every cell of the grid corresponds to a specific downsampling method $\eta$ (row), disease $\mathcal{M}$ (column). Within each cell, there are five lines with distinct colors; each color denotes a corresponding underlying population $V$. Each colored line is formed by sequentially connecting adjacent points with straight lines. There are six points for each colored line, each associated with a sample interval in the order of $\xi_+ = 10, 30, 60, 90, 180, 360$, denoting the KL-divergence of the baseline $\boldsymbol{\rho}(\mathcal{M}, \mathcal{D}_{\xi_0, \eta})$ from $\boldsymbol{\rho}(\mathcal{M}, \mathcal{D}_{\xi_+, \eta})$ with the sample interval $\xi_+$.

with an increasing degree of dissimilarity. We found `Upperbound` is better at preserving likely paths than `Snapshot`, and the limits of the sampling interval needed to preserve likely paths of disease spreading lies amongst $\xi \in \{10, 30, 60\}$. Under `Upperbound`, diseases with similar $R_0$ resemble each other, and MERS with a low $R_0 = 0.69$, has its likely paths varying notably over sampling intervals for a less "diffuse" population, while other diseases—despite exhibiting a wide range of $R_0 \in [0.69, 15]$—maintain a similar pattern of those likely paths with rising sampling interval, for a given population.

**Kullback-Leibler divergence on individual infection risk.**    As shown in Fig 10, the `Snapshot` method in general will exhibit higher divergence than the `Upperbound` method, except for diseases with low $R_0$, such as influenza type A (1.31). In general, the higher the $\delta_{\xi_+}$, the higher the divergence of individual infection risk from estimations with $\xi_+$-downsampled contact data when compared to $\xi_0$-sampled contact data.

## Discussion

We investigated the impacts of observation frequency on proximity contact data and modeled transmission dynamics through the downsampling-simulation-evaluation approach. We started by specifying various sensing regimes regarding the sampling method and the sampling interval. This exploration drew on two sampling methods proposed here—the `Snapshot` method as a conceptually and practically straightforward scheme and the `Upperbound`

method as a theoretical upper bound. We then emulated proximity contact data collected with different sensing regimes by downsampling from a baseline. Finally, we designed experiments and metrics to evaluate the impacts of sensing regimes on proximity-contact-data-informed transmission models under various circumstances.

We evaluated the impacts of sensing regimes on transmission dynamics from both population-level and individual-level perspectives under the circumstances of different underlying populations and types of diseases. In summary, our results show that:

- In general, the sensing regime impacts multiple metrics of proximity-contact-data-informed transmission models, at both the population level and the individual level.

- The impact of sensing regimes varies notably as one varies the underlying population and disease type.

- As the sampling interval increases, proximity contact data downsampledwith both the `Snapshot` method and the `Upperbound` method leads the informed ABM-SEIR model to diverge from the baseline.

- For underlying populations captured by the SHED datasets and the types of diseases that we investigated, it appears that we need a sampling interval of less than 30 minutes to mitigate the risk of having downsampled proximity contact data misrepresenting the baseline. For example, the shapes of the violin plots shifted when the sampling interval increases over 30 minutes in Fig 1, with the `Snapshot` method, under the circumstance of SHED8-measles and in almost every cell of the corresponding `Upperbound` grid, as shown in Fig 2.

- As the sampling interval increases, the `Snapshot` method tends to retain reasonable estimations in general, but bears the risk of underestimating the spread of diseases compared with the baseline. For example, the `Snapshot` method may result in differences for metrics such as cumulative cases (Figs 1 and 3–6), attack rate (Fig 7), and infection period (Figs 3–6), especially under the circumstance of a "close" community like SHED1 and a high $R_0$ disease like measles. In contrast, the `Upperbound` method tends to systematically overestimate population-level metrics compared with the baseline (Figs 2–6 and 8). However, the `Upperbound` method can retain good estimations on the period of disease spread for high $R_0$ disease (Figs 2 and 5 and for individual infection risks (Fig 10).

- The results of downsampling with the `Snapshot` method exhibit higher discrepancy from the baseline under circumstances of a "close" community for a high $R_0$ disease than for a "diffuse" community with a low $R_0$ disease (Figs 7 and 10). Such tradeoffs raise the potential for improving sampling by creating a hybrid scheme that draws on the best of the `Snapshot` and the `Upperbound` methods.

- For parameters that depict a disease, the $R_0$ seems to be important in determining the impacts of sampling frequency and sampling method on the results of transmission models. The importance of $R_0$ is evident by comparing MERS with pertussis, as mentioned above. Such importance is also apparent when comparing COVID-19 with its variants, which are characterized here by an identical latent period and infectious period, as shown in the S1 Appendix. Besides the $R_0$, we have yet to find notable impacts from the latent period and the infectious period when we compare pairs of diseases with similar $R_0$ but different latent periods and infectious periods, such as measles and chickenpox, as shown in the S1 Appendix.

To the best of our knowledge, we are among the first to characterize how temporal resolution of proximity contact data will impact results from an agent-based transmission model. We found that to secure reliable results from the ABM-SEIR model examined here, it will

generally be sufficient to sample proximate contacts with the `Snapshot` method using an inter-sample interval of a half-hour or below. Furthermore, we noticed that the most effective sampling method and sampling interval can depend on the types of disease. Among the combinations of the underlying populations and types of diseases we have investigated, we further found that the most favorable sensing regime may differ depending on the population-level and individual-level metrics of interest.

Our methodology for investigating the impact of the sensing regime for proximity contact data on results of informed transmission models is highly generalizable. Such methodology can be adapted to tailor sensing regimes for specific circumstances and evidence the study of proximity contact networks for transmission modeling.

Our findings are subject to a number of important limitations:

- **Limited Population Size**—Given the confined population size, it is possible that the observed behavior of `Snapshot` and `Upperbound` here is materially altered by quantization effects exhibited by agent-based simulation when population size is small [97–99]. Because the attack rate represents the quotient of two integers, when both nominator and denominator are small, the possible values of their ratio can be sparse. A partial result is that realizations in which the entire population is infected can happen more frequently than when the population size is large.

- **Limited Diversity of Participants**—As university students, most participants of our experiments share similar lifestyles and activity spaces for their working and studying hours. Because network structures of proximate contact in different age groups and communities can be a key factor in disease transmission [100], the findings resulting from applying such methods to larger and more societally representative communities may vary notably from the results shown here. Future work is encouraged to adapt our methodology for a broader collection of underlying populations and types of diseases.

- **Simple Modeling Methodologies**—While the method of feeding high-resolution proximity contact data into the epidemiological model used in our experiment is mathematically and practically straightforward, there is much opportunity to apply more versatile methods to combine data into the model as modeling methodologies advance. Our experiment is designed to examine the impact of observation frequency on transmission models informed by proximity contact data. Future studies that collect longitudinal incidence reports along with proximate contacts are needed to enable evaluations of the performance of transmission models informed by proximity contact data in predicting real-world outbreaks.

In summary, this work has investigated the impacts of observation frequency and sampling methods for proximity contact records in capturing proximity contact networks for epidemiological simulations. We evaluated the impacts induced by the temporal granularity of sampling networks in terms of distortion of measured network structure and of population-level and individual-level simulation outcome metrics in light of combinations of specific diseases and underlying types of communities. These results emphasize classes of pathogens and population structures in which the design of new studies should prioritize frequent sampling of contact networks. Our findings also provide guidance as to how network density and lower sampling rates might distort measures such as attack rate and individual risk.

## Supporting information

**S1 Table. Welch's *t*-tests over observation intervals.** It contains two comma-separated values (CSV) files, one for each of the `Snapshot` and the `Upperbound` downsampling method.

Each CSV file contains details of the corresponding Welch's *t*-test, where block names are underscore-concatenated strings of the underlying population and the type of disease/pathogens.
(ZIP)

**S1 Appendix. Auxiliary results and model diagrams.** Auxiliary results for 12 diseases, as well as a diagram illustrating the `SEIR-ABM`.
(PDF)

## Acknowledgments

We would like to thank Greg Oster and Mohammad Hashemian for technical consultant and help. NDO wants to extend special gratitude to SYK for inspiring this work and support throughout the research period.

## Author Contributions

**Conceptualization:** Weicheng Qian, Kevin Gordon Stanley, Nathaniel David Osgood.

**Data curation:** Kevin Gordon Stanley, Nathaniel David Osgood.

**Formal analysis:** Weicheng Qian, Kevin Gordon Stanley, Nathaniel David Osgood.

**Funding acquisition:** Kevin Gordon Stanley, Nathaniel David Osgood.

**Investigation:** Weicheng Qian, Kevin Gordon Stanley, Nathaniel David Osgood.

**Methodology:** Weicheng Qian, Kevin Gordon Stanley, Nathaniel David Osgood.

**Project administration:** Kevin Gordon Stanley, Nathaniel David Osgood.

**Resources:** Kevin Gordon Stanley, Nathaniel David Osgood.

**Software:** Weicheng Qian.

**Supervision:** Kevin Gordon Stanley, Nathaniel David Osgood.

**Validation:** Kevin Gordon Stanley, Nathaniel David Osgood.

**Visualization:** Weicheng Qian.

**Writing – original draft:** Weicheng Qian, Kevin Gordon Stanley, Nathaniel David Osgood.

**Writing – review & editing:** Weicheng Qian, Kevin Gordon Stanley, Nathaniel David Osgood.

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
