## [Decision Letter · Decision Letter 0]

11 Jul 2022

Dear Mr Qian,

Thank you very much for submitting your manuscript "Impacts of observation frequency on proximity contact data and modeled transmission dynamics" for consideration at PLOS Computational Biology.

As with all papers reviewed by the journal, your manuscript was reviewed by members of the editorial board and by several independent reviewers. In light of the reviews (below this email), we would like to invite the resubmission of a significantly-revised version that takes into account the reviewers' comments.

I agree with the reviewer comments that this is potentially a valuable contribution and that its focus on combining social and mobility data is a meaningful advance. While I agree with the Reviewer comment that comparing to empirical data would strengthen the study, but I think there's a path forward without necessarily adding in an empirical comparison. I will leave that decision up to the authors. However, I think a bigger concern is around the data sharing. There’s already a lot of work on longitudinal contact networks and they almost always provide more granular data. Because the actual model and results are not in and of themselves novel, the value of this paper is strongly driven by the data themselves. That you can only share aggregated data because of what the participants agreed is a study design issue and significantly lowers the future value of the study. I appreciate that it's harder work to share longitudinal data, but SocioPatterns has done it, Salute has done it, and researchers like Sune Lehmann have done it. Can the authors provide any mechanism for future authors to obtain access to the more granular data (in this case I mean beyond simply saying, "available upon reasonable request). If the authors cannot address this data availability issue, they should provide strong justification for how this advances from existing work on longitudinal studies without providing access to the data.

We cannot make any decision about publication until we have seen the revised manuscript and your response to the reviewers' comments. Your revised manuscript is also likely to be sent to reviewers for further evaluation.

Sincerely,

Samuel V. Scarpino

Associate Editor

PLOS Computational Biology

Virginia Pitzer

Deputy Editor-in-Chief

PLOS Computational Biology

I agree with the reviewer comments that this is potentially a valuable contribution and that its focus on combining social and mobility data is a meaningful advance. While I agree with the Reviewer comment that comparing to empirical data would strengthen the study, but I think there's a path forward without necessarily adding in an empirical comparison. I will leave that decision up to the authors. However, I think a bigger concern is around the data sharing. There’s already a lot of work on longitudinal contact networks and they almost always provide more granular data. Because the actual model and results are not in and of themselves novel, the value of this paper is strongly driven by the data themselves. That you can only share aggregated data because of what the participants agreed is a study design issue and significantly lowers the future value of the study. I appreciate that it's harder work to share longitudinal data, but SocioPatterns has done it, Salute has done it, and researchers like Sune Lehmann have done it. Can the authors provide any mechanism for future authors to obtain access to the more granular data (in this case I mean beyond simply saying, "available upon reasonable request). If the authors cannot address this data availability issue, they should provide strong justification for how this advances from existing work on longitudinal studies without providing access to the data.

Reviewer's Responses to Questions

**Comments to the Authors:**

Reviewer #1: Review is uploaded as an attachment.

Reviewer #2: Comments

The authors present a useful and practical work for contact tracing spread of infectious diseases, especially while the COVID-19 pandemic.

There are many countries started to use auto contact tracing apps via smartphones.

This study is an interesting work to evaluate the data sampling approaches and is helpful in preparation for potential outbreaks in the future.

The study consists of several important components in infectious disease modeling such as contact network and agent-based SEIR model, and practically uses individual traced data to investigate the impact of the network and model given 2 downsampling methods with lower sampling frequencies.

The authors apply the 5 sampled networks on 12 diseases and compared 2 downsampling methods with 7 frequencies (1 baseline and 6 alternatives).

- Even though the sampling population could be too small and be biased due to its closed population in the university but this is still useful to demonstrate the methodology of the study.

- Overall, the manuscript is written nicely and some comments are listed in the followings.

- recommended study about contact tracing app in the UK: [Wymant, C., Ferretti, L., Tsallis, D. et al. The epidemiological impact of the NHS COVID-19 app. Nature 594, 408–412 (2021). https://doi.org/10.1038/s41586-021-03606-z]

The introduction of the manuscript is smoothly written and describes the objectives with three research questions clearly.

- Page 3/30: The authors can add a citation of Table 1 while mentioning the reproduction numbers.

- It would be recommended to answer these questions more systematically in Discussion.

The methods and results need to be organized better and more clear description or systematic flow is needed for readers to understand better without going back and forth between Methods and Results.

- The authors can consider moving some part of the Methods into the main text or/and to add more description for example while explaining the two downsampling methods in 1st paragraph of the results.

- Table 1 is a good summary of all 12 diseases in this study. Adding a short paragraph in Method to describe the diseases (such as Fifth disease caused by Parvovirus B19 and much more common among children than adult) would be recommended. This type of information is important while applying the methodology using data collected in university. For discussion, the network structures in different age groups and communities (household vs school) could be a key factor in transmission. The pathogen or variant types can be mentioned there. "disease/pathogen" can be replaced by simple "disease".

- Table 1: 2nd row, replace "COVID-19" by "COVID-19 wild type".

Cumulative cases and ECDFs

- It is quite difficult to read all the figures especially figures 1 to 4. There is too much information and the font is too small. The authors can select some of the presented results (such as the section of "Outbreaks and outbreak timing") and put the others in the supplementary. Also, the figures 1 and 2 can be combined in one and so for figure 3 and 4 (for example using color and shape for sampling interval and downsampling method, respectively).

- The section of "Outbreaks and outbreak timing" can be part of "Cumulative cases". In figure 5a, it is clear to see that Upperbound method (blue line) slow down the spread while interval increases, but all methods seem to behavior the same in the bottom-left panel. Can you explain this? Would this be related to the stochasticity of the agent-based model? In the description on Page 21/30, 30 simulations per scenario seem to be too low, but I can understand that the output data size is quite large, 85GB. In Figure 5c, while comparing by columns (for example SHED 7 and 9, both diffuse communities), one tends to slow down and one tends to speed up the spread. Would this happen because of the biases of the sampled network data?

- On Page 5/30: the Xi symbol is not defined until Page 7/30.

- Figures 1 and 2: what are the blue lines and red dots? More detailed captions in all figures are needed.

- The two supplementary spreadsheets might be uploaded as one excel file.

- Page 10/30: (ECDF) is written twice.

Attack rates

- On Page 8/30, Line 5: Should that be the curvy V and normal V in {D, V, M}? The initial infectious individual (curvy V) is not mentioned until the 2nd paragraph.

- The concept of the two V is described in Methods but it is a bit confusing while reading this section.

Infection pairs

- Page 12/30: In the first paragraph, a better description of the resulting values of Weighted Minkowski Distance (Figure 6) and KL divergence (Figure 7) is needed, especially while grouping two types of diseases. Would that be associated with the reproduction numbers, as mentioned on Page 13/30? Or also the incubation periods and infectious periods listed in Table 1? One can also investigate how these three factors affecting the results, for example, all 4 COVID-19 variants share the same incubation periods and infectious periods and the only variable is the R0. This affects the results in Figure 3 but not Figure 4. On the other hand, long and short infectious periods can be compared using pertussis vs measles given both sharing similar R0 and incubation period.

- Page 12/30: Should the darker/lighter color refers to green/red in Figure 6?

- Figure 6: what is the x-y axis?

Individual infection risks

- Page 13/30, is the (Xi_+) symbol the same as (Xi') symbol on page 8/30?

Reviewer #3: In their paper, the authors present a new method to better combine social information from cellular networks with an epidemiological model to model the spread of various infectious diseases. Indeed, over the past two years, there has been an increased interest in accurately creating models to predict the spread of epidemics, so many researchers across different fields have worked to develop models that can faithfully predict plague outbreaks under various scenarios.

However, this article does not provide enough evidence that an ABM-SEIR, which uses data from mobile devices, is the best method of tracking pandemic spread. Hence, this paper should focus on fewer pathogens and primarily show how this model can be used to predict an outbreak of COVID-19 in contrast to the SEIR model (or another model)l, which doesn't use such external information.

In addition, I have several minor comments regarding the presentation of the data in the paper:

While the authors present an innovative method for collecting data that can be used in epidemiological ABM, the paper itself is hard to read, and all the figures are not self-explanatory because the data they contain is so large. Although the authors attempted to include as many pathogens as possible, this resulted in a detailed paper that could have been better organized. Since each calculation contains five data sets, the authors should only present three pathogens per figure. By doing so, the figures will be more readable, and readers will better understand the differences between the different data sets and methods employed in this study. Hence, I think this paper can be better organized, and the figures have to be changed to contain less information (the original figures can use as supplementary material).

**Have the authors made all data and (if applicable) computational code underlying the findings in their manuscript fully available?**

Reviewer #1: **No: **They cannot make participants' data (SHEDs) accessible to the public

because our ethics board has concerns. Such concerns include that no safety

measures can ensure participants' privacy for public access because SHEDs'

proximate contact data is longitudinal (over a month) and with high granularity

(collected every 5 minutes). Additionally, in the informed consent form that the

participants signed, researchers committed to only providing data in aggregate to

ensure privacy. Therefore, to disclose the full dataset would violate that commitment.

Reviewer #2: **No: **the code is available online but not the data. summary statistics in supplementary is needed.

Reviewer #3: Yes

PLOS authors have the option to publish the peer review history of their article (what does this mean?). If published, this will include your full peer review and any attached files.

Reviewer #1: No

Reviewer #2: **Yes: **Louis Yat Hin Chan

Reviewer #3: No
---

## [Decision Letter · Decision Letter 1]

1 Dec 2022

Dear Mr Qian,

Thank you very much for submitting your manuscript "Impacts of observation frequency on proximity contact data and modeled transmission dynamics" for consideration at PLOS Computational Biology. As with all papers reviewed by the journal, your manuscript was reviewed by members of the editorial board and by several independent reviewers. The reviewers appreciated the attention to an important topic. Based on the reviews, we are likely to accept this manuscript for publication, providing that you modify the manuscript according to the review recommendations.

The authors have done a nice job with the revision. However, I still would like to see more in terms of data availability. In their response, the authors mention that aggregate data can be made available. In many cases, those aggregate data are likely to be useful for epidemic simulations. Can the authors provide aggregate metrics, e.g., degree distributions over time, measure of network structure like cluster over time, etc.?

Sincerely,

Samuel V. Scarpino

Academic Editor

PLOS Computational Biology

Virginia Pitzer

Section Editor

PLOS Computational Biology

The authors have done a nice job with the revision. However, I still would like to see more in terms of data availability. In their response, the authors mention that aggregate data can be made available. In many cases, those aggregate data are likely to be useful for epidemic simulations. Can the authors provide aggregate metrics, e.g., degree distributions over time, measure of network structure like cluster over time, etc.?

Reviewer's Responses to Questions

**Comments to the Authors:**

Reviewer #2: Comments

There are lots of places improved in this revised version. The figures are simpler and easier for readers to understand where to focus on. The results of using NECC show consistence between scenarios.

The article is more organized and readable by placing Results after Methods. However, most of the 1st paragraphs in subsections of Results should be placed in Methods to explain and define all symbols and notations. For example, Lines 475-502 on Page 14/31 should be Methods on Page 10/31 (note that eta is not defined right after first presented on Line 482. It can be mentioned on Page 6 or 7). The same on Lines 540-549 on Page 15/31 should be Methods on Page 10/31. Lines 615-627 on Page 20/31 are also Methods where notation eta and xi_+ should be predefined.

Some minors:

- 'basic reproductive number' instead of 'base reproductive number' on Line 246 Page 7/31.

- Line 637: delete 'two'.

- COVID-19 wild type is added in the table but not in the main text for example on page 5 and 7.

- While mentioning the time interval between sampling t_i in [Xi i, Xi i + Xi), it can be written as i Xi, (i+1) Xi to avoid mixing up with index i on for example Line 174 on Page 5/31.

Reviewer #3: As a result of the authors' revisions, the article has been improved.

Having the methods part appear prior to the results does improve understanding of the article and simplifies the paper. Moreover, the figures are presented better in the current version of the article than in the original (Fig. 2 is still unreadable).

However, In my opinion, an article presenting an epidemiological model for predicting the rate of pandemic spread must be compared to the actual data in some way. In the end, epidemiological models are designed to monitor the spread of a pandemic and to facilitate informed decision-making. The fact that after two and a half years, the coronavirus still exists to a different extent in most countries of the world shows us that we should not take the global pandemic lightly, which emphasizes the necessity for epidemiological models that are as precise as possible. As a result of frequent changes in how the Corona pandemic spreads (for example, the introduction of new variants and the effectiveness of vaccinations against infection/severe morbidity), and on the other hand, there has been a change in epidemic control policy (i.e., there are almost no lockdowns in the world as compared to two years ago), most models produced mediocre results in predicting the spread of the pandemic and have been criticized regularly. Therefore, I find that an article that ultimately presents an epidemiological model for predicting the rate of spread of an epidemic must be compared in one way or another to the real data. Furthermore, it would be necessary to convince the reader that this model, which uses cellular data, has an advantage over epidemiological models that do not utilize this information because the information required for this model is not publicly available in all countries.

As a result, it is the editor's decision whether to publish the article in this format.

**Have the authors made all data and (if applicable) computational code underlying the findings in their manuscript fully available?**

Reviewer #2: **No: **the authors explained in the reply.

Reviewer #3: **No: **Data is only available upon request. Using this data requires an application to the institution's research ethics board. The editor's role is to decide whether this level of data sharing is acceptable.

PLOS authors have the option to publish the peer review history of their article (what does this mean?). If published, this will include your full peer review and any attached files.

Reviewer #2: **Yes: **Louis Yat Hin Chan

Reviewer #3: No

Figure Files:

Data Requirements:

Reproducibility:

References:

---

## [Editor Report · Decision Letter 2]

3 Feb 2023

Dear Mr Qian,

We are pleased to inform you that your manuscript 'Impacts of observation frequency on proximity contact data and modeled transmission dynamics' has been provisionally accepted for publication in PLOS Computational Biology.

Best regards,

Samuel V. Scarpino

Academic Editor

PLOS Computational Biology

Virginia Pitzer

Section Editor

PLOS Computational Biology

---

## [Editor Report · Acceptance letter]

21 Feb 2023

PCOMPBIOL-D-22-00711R2 

Impacts of observation frequency on proximity contact data and modeled transmission dynamics

Dear Dr Qian,

I am pleased to inform you that your manuscript has been formally accepted for publication in PLOS Computational Biology. Your manuscript is now with our production department and you will be notified of the publication date in due course.

With kind regards,

Anita Estes
